# Immunogenicity and reactogenicity of SARS-CoV-2 vaccines BNT162b2 and CoronaVac in healthy adolescents

Jaime S. Rosa Duque [1,10], Xiwei Wang[1,10], Daniel Leung[1,10], Samuel M. S. Cheng [2,10], Carolyn A. Cohen[2,3,10], Xiaofeng Mu[1], Asmaa Hachim[2,3,4], Yanmei Zhang[1], Sau Man Chan[1], Sara Chaothai[2], Kelvin K. H. Kwan[2], Karl C. K. Chan[2], John K. C. Li[2], Leo L. H. Luk[2], Leo C. H. Tsang[2], Wilfred H. S. Wong[1], Cheuk Hei Cheang[1], Timothy K. Hung [1], Jennifer H. Y. Lam[1], Gilbert T. Chua[1], Winnie W. Y. Tso[1,5], Patrick Ip[1], Masashi Mori[6], Niloufar Kavian[2,3,7], Wing Hang Leung[1], Sophie Valkenburg [2,3,8✉], Malik Peiris [2,9✉], Wenwei Tu [1✉] & Yu Lung Lau [1✉]

We present an interim analysis of a registered clinical study (NCT04800133) to establish immunobridging with various antibody and cellular immunity markers and to compare the immunogenicity and reactogenicity of 2-dose BNT162b2 and CoronaVac in healthy adolescents as primary objectives. One-dose BNT162b2, recommended in some localities for risk reduction of myocarditis, is also assessed. Antibodies and T cell immune responses are non-inferior or similar in adolescents receiving 2 doses of BNT162b2 (BB, $N = 116$) and CoronaVac (CC, $N = 123$) versus adults after 2 doses of the same vaccine (BB, $N = 147$; CC, $N = 141$) but not in adolescents after 1-dose BNT162b2 (B, $N = 116$). CC induces SARS-CoV-2 N and N C-terminal domain seropositivity in a higher proportion of adolescents than adults. Adverse reactions are mostly mild for both vaccines and more frequent for BNT162b2. We find higher S, neutralising, avidity and Fc receptor-binding antibody responses in adolescents receiving BB than CC, and a similar induction of strong S-specific T cells by the 2 vaccines, in addition to N- and M-specific T cells induced by CoronaVac but not BNT162b2, possibly implying differential durability and cross-variant protection by BNT162b2 and CoronaVac, the 2 most used SARS-CoV-2 vaccines worldwide. Our results support the use of both vaccines in adolescents.

[1] Department of Paediatrics and Adolescent Medicine, The University of Hong Kong, Hong Kong, China. [2] School of Public Health, The University of Hong Kong, Hong Kong, China. [3] HKU-Pasteur Research Pole, School of Public Health, The University of Hong Kong, Hong Kong, China. [4] Sir William Dunn School of Pathology, University of Oxford, Oxford, UK. [5] State Key Laboratory of Brain and Cognitive Sciences, The University of Hong Kong, Hong Kong, China. [6] Research Institute for Bioresources and Biotechnology, Ishikawa Prefectural University, Nonoichi, Japan. [7] IRCCS, Humanitas Research Hospital, Milan, Italy. [8] Department of Microbiology and Immunology, Peter Doherty Institute for Infection and Immunity, University of Melbourne, Melbourne, Australia. [9] Centre for Immunology and Infection, Hong Kong, China. [10]These authors contributed equally: Jaime S. Rosa Duque, Xiwei Wang, Daniel Leung, Samuel M. S. Cheng, Carolyn A. Cohen. ✉email: sophie.v@unimelb.edu.au; malik@hku.hk; wwtu@hku.hk; lauylung@hku.hk

The coronavirus disease 2019 (COVID-19) pandemic due to the severe acute respiratory syndrome coronavirus 2 (SARS-CoV-2) continues to cause significant morbidity, mortality and socioeconomic disruptions worldwide[1]. While acute COVID-19 in children results in fewer hospitalisations and deaths than adults, they can lead to serious complications, such as multisystem inflammatory syndrome in children (MIS-C)[2]. COVID-19 also has profound negative impact on school attendance, neurodevelopment and mental health in the paediatric population[3,4]. Several vaccines against SARS-CoV-2, utilising novel nucleoside-modified mRNA technologies and the conventional inactivated whole-virus platform, underwent an expeditious review process by the World Health Organization (WHO) and were deemed adequately safe and efficacious for emergency use in adults during the initial phase[5,6]. Landmark phase 3 clinical trials reported efficacies of ~90–95% for the mRNA vaccine, BNT162b2, and ~50–85% for the inactivated whole-virus vaccine, CoronaVac, against symptomatic COVID-19 in persons aged ≥16 and ≥18 years old, respectively[7,8]. The vaccine efficacy for 12 to 15-year-olds receiving BNT162b2 was 100% in a phase 3 study[9]. Another phase 2 trial demonstrated that the seroconversion rate for 2 doses of CoronaVac in ages 12–17 years old was 100% as well[10].

Vaccine efficacy has been linked to markers of immunological response known as correlates of protection (COP) in many infectious diseases, most commonly neutralising antibody titres[11–14]. However, host defence against the viral infection involves many constituents of the immune system acting synergistically and dynamically, rather than merely reflected by antibody neutralisation[15,16]. As examples, spike protein (S) IgG are also a COP against symptomatic COVID-19, and the onset of efficacy in mRNA vaccines coincides with the presence of binding antibodies and precedes neutralising antibody production[17,18]. Optimised antibody avidity and Fc receptor-binding are also implicated in superior potency of antibody cocktail treatments for COVID-19[19]. Additionally, non-S SARS-CoV-2 structural proteins, such as the nucleocapsid (N) and membrane (M), are associated with antibody responses in convalescent patients, and in fact, the C-terminal domain of N (N-CTD) is more specific to SARS-CoV-2[20–22]. Therefore, studies on immunogenicity outcomes that include these components are necessary.

To prevent progression to severe illness, T cells also play a major role in orchestrating a focused spectrum of immune responses, such as directing apoptosis of infected cells and antibody germinal centre reactions for high-avidity class-switched responses[18,23–25]. This is evident from studies that show humans who have inborn errors of immunity affecting T cells suffer from more severe and fatal viral infections, including COVID-19[26]. Most studies that investigated cellular immunogenicity focused on S-specific T cell responses, but non-spike proteins are relatively conserved and immunodominant for T cell responses in natural infection[27,28]. Two recent studies also hinted at a role of pre-existing cross-reactive T cells aborting SARS-CoV2 infections by examining frequently tested healthcare workers or household contacts who remained PCR-negative[29,30]. Unfortunately, COPs remain difficult to define due to the types of specimens required for assessment, workload in sample preparation and the functional complexity of assays. Characterising all these humoral and cellular constituents against S and non-spike proteins collectively in addition to neutralising antibody titres alone are essential for our understanding about vaccine responses to whole-virus vaccines but were not included in previous studies. In a recent head-to-head evaluation, BNT162b2 induced higher neutralisation antibody levels, avidity and Fc receptor-binding antibodies in healthy adults, while T cell responses against SARS-CoV2 structural proteins were greater for those who received CoronaVac, which also contains other structural proteins in addition to S[31].

Achieving a comprehensive understanding of vaccine-induced humoral and cellular immune responses is important for future development and approval of novel immunisation platforms and boosters. Comprehensive comparative immunogenicity analyses of different vaccines allow us to investigate the contribution of different arms of the immune system to vaccine efficacy. These studies are rare in younger age group.

Additionally, clinical trials and post-marketing surveillance reported adverse events (AEs), such as hypersensitivity reactions and Bell's palsy, which have sparked public concerns[32,33]. In adolescents, our group amongst others recently found an increased incidence of myocarditis/pericarditis as high as 1 in 3,000 second doses of BNT162b2 in male adolescents, prompting Hong Kong (HK) and the UK to recommend a single dose of BNT162b2 for adolescents only[34–36]. While mRNA vaccines are linked to frequent systemic adverse reactions (ARs), reactogenicity appeared to be milder for the inactivated COVID-19 vaccine.

In this study, we aimed to perform an immunobridging study showing the humoral and cellular immunogenicity in adolescents receiving 1 and 2 doses of BNT162b2 and 2 doses of CoronaVac are non-inferior to adults, especially to inform on the use of CoronaVac in children for which there are no efficacy and effectiveness data at the time of writing. We compared various humoral and cellular response outcomes in adolescents to BNT162b2 and CoronaVac head-to-head, which are the top 2 most used COVID-19 vaccines in the world[37]. This paper presents a pre-specified interim analysis of the immunogenicity, reactogenicity and safety results at 1 month after 2 doses of the COVID-19 vaccine.

## Results

**Enrolment of study participants.** A total 658 volunteers were screened, of whom 646 provided consent, consisting of 309 adolescents and 337 adults at dose 1, respectively, and were enroled between 27 April 2021 and 23 October 2021 (see Methods; Supp. Fig. 1). Based on clinical history and serological screening, 26 were enroled in separate prior COVID-19 and 93 in immune/paediatric diseases sub-studies. This presents interim analysis of our study that will track severe adverse events and immunogenicity outcomes over a 3-year period focused on healthy participants, consisting of 239 adolescent (11–17 years old, mean = 14.0, SD = 1.7) and 288 adult (18–67 years old, mean = 47.5, SD = 7.5) participants (total $N = 527$) who received at least 1 dose of either BNT162b2 or CoronaVac in the healthy safety population (for reactogenicity and safety analyses see Methods, and Protocol and Statistical Analysis Plan in Supplementary Information). Adolescent participants all returned for a subsequent follow-up visit (Visit 2) and completed the 2-dose (BB for BNT162b2 and CC for CoronaVac) vaccination series. Adolescent and adult participants who completed 2 doses returned for the subsequent follow-up visit (Visit 3). Demographic characteristics (Supp. Table 1) were evenly distributed. The evaluable analysis population in this analysis included those uninfected as assessed at any study visits, with no major protocol deviations and had a valid immunogenicity result (see Methods; Supp. Fig. 1). There were 223 adolescents and 166 adults in the evaluable analysis population for primary immunogenicity after 2 doses. We confirmed the findings using a secondary analysis in an expanded analysis population, consisting of 226 adolescents and 223 adults that had relaxed vaccination and blood taking intervals (see Methods; Supp. Fig. 1).

**Table 1 Humoral immunogenicity outcomes in evaluable analysis populations by vaccine regimen.**

| | BNT162b2 | | | CoronaVac | |
| --- | --- | --- | --- | --- | --- |
| | Adolescents | Adolescents | Adults | Adolescents | Adults |
| | 1 dose | 2 doses | 2 doses | 2 doses | 2 doses |
| **Antibody responses** | | | | | |
| **S IgG on ELISA** | | | | | |
| N | 101 | 103 | 115 | 116 | 50 |
| GM OD450 value (95% CI) | 0.53 (0.47-0.60) | 1.21 (1.17-1.25) | 1.11 (1.07-1.16) | 0.54 (0.49-0.58) | 0.42 (0.36-0.50) |
| % positive (>/ = LOD at 0.3) | 100%, $P > 0.9999$ | 100%, $P > 0.9999$ | 99.1% | 94.0%, $P = 0.0228$ | 82.0% |
| **S-RBD IgG on ELISA** | | | | | |
| N | 107 | 104 | 115 | 119 | 51 |
| GM OD450 value (95% CI) | 1.96 (1.83-2.09) | 2.64 (2.53-2.75) | 2.73 (2.63-2.83) | 1.20 (1.10-1.31) | 1.20 (1.04-1.37) |
| % positive (>/ = LOD at 0.5) | 100%, $P > 0.9999$ | 100%, $P > 0.9999$ | 100% | 96.6%, $P > 0.9999$ | 96.1% |
| **S-RBD ACE2-blocking antibody on sVNT** | | | | | |
| N | 107 | 104 | 115 | 119 | 51 |
| GM % inhibition (95% CI) | 81.3% (79.2-83.5%) | 97.1% (97.0-97.2%) | 94.9% (94.3-95.5%) | 71.2% (66.7-76.0%) | 54.6% (48.5-61.4%) |
| % positive (>/ = LOQ at 30%) | 100%, $P > 0.9999$ | 100%, $P > 0.9999$ | 100% | 96.6%, $P = 0.43$ | 94.1% |
| **Neutralising antibody on PRNT** | | | | | |
| N | 63 | 60 | 13 | 64 | 19 |
| GM PRNT90 (95% CI) | 14.4 (11.9-17.4) | 115 (93.3-140) | 64.6 (43.5-96.1) | 9.58 (8.50-10.8) | 7.75 (6.35-9.46) |
| % positive (>/ = LOD at 10) | 85.7%, $P = 0.34$ | 100%, $P > 0.9999$ | 100% | 75.0%, $P = 0.16$ | 57.9% |
| GM PRNT50 (95% CI) | 45.2 (36.1-56.5) | 331 (277-396) | 259 (168-398) | 28.0 (23.9-32.8) | 21.5 (15.4-30.0) |
| % positive (>/ = LOD at 10) | 98.4%, $P > 0.9999$ | 100%, $P > 0.9999$ | 100% | 100%, $P = 0.23$ | 94.7% |
| **S IgG avidity on ELISA** | | | | | |
| N | 88 | 103 | 114 | 109 | 41 |
| GM avidity index (95% CI) | 21.5% (19.5-23.8) | 29.7% (27.9-31.5) | 23.5% (22.0-25.1) | 20.5% (19.1-22.1) | 12.0% (10.8-13.3) |
| **S IgG FcγRIIIa-binding on ELISA** | | | | | |
| N | 101 | 103 | 115 | 116 | 50 |
| GM OD450 value (95% CI) | 1.12 (1.00-1.26) | 2.07 (2.02-2.11) | 1.93 (1.87-1.99) | 0.75 (0.65-0.86) | 0.60 (0.48-0.74) |
| % positive (>/ = LOD at 0.28) | 100%, $P > 0.9999$ | 100%, $P > 0.9999$ | 100% | 87.1%, $P = 0.81$ | 86.0% |
| **N IgG on ELISA** | | | | | |
| N | / | / | / | 64 | 21 |
| GM OD450 value (95% CI) | / | / | / | 1.72 (1.61-1.83) | 0.77 (0.59-0.99) |
| % positive (>/ = LOD at 0.88) | / | / | / | 98.4%, $P < 0.0001$ | 52.4% |
| **N-CTD IgG on ELISA** | | | | | |
| N | / | / | / | 64 | 21 |
| GM OD450 value (95% CI) | / | / | / | 2.09 (1.89-2.31) | 0.92 (0.73-1.17) |
| % positive (>/ = LOD at 1.34) | / | / | / | 92.2%, $P < 0.0001$ | 28.6% |

*GM* geometric mean, *OD* optical density, *LOD* limit of detection, *LOQ* limit of quantification, *CI* confidence interval, *S* spike protein, *ELISA* enzyme-linked immunosorbent assay, *RBD* receptor-binding domain, *ACE-2* angiotensin-converting enzyme-2, *sVNT* surrogate virus neutralisation test, *PRNT* plaque reduction neutralisation test, *PRNT90* 90% plaque reduction neutralisation titre, *PRNT50* 50% plaque reduction neutralisation titre, *FcγRIIIa* Fc gamma receptor III-a, *N* nucleocapsid protein, *CTD* C-terminal domain, *IFN-γ* interferon-gamma, *IL-2* interleukin-2.
*P*-values compare the proportion of positive responses between adolescents receiving 1 or 2 doses of vaccine and adults receiving 2 doses of the same vaccine by two-tailed Fisher exact test.

**Humoral immunogenicity outcomes between adolescents and adults.** For the primary humoral immunogenicity analysis, SARS-CoV-2 S IgG, S-RBD IgG by enzyme-linked immunosorbent assay (ELISA), surrogate virus neutralisation test (sVNT), plaque reduction neutralisation test (PRNT), S IgG avidity and S IgG Fcγ receptor IIIa (FcγRIIIa)-binding on ELISA were performed for healthy, uninfected adolescents who received BB or CC (see Methods). Since there had been an interim recommendation to vaccinate adolescents with only 1 dose of BNT162b2 as the primary series in HK and the UK due to the higher risk of myocarditis after 2 doses, we also tested whether adolescent B was non-inferior. Evaluable adolescent BNT162b2 recipients achieved 100% S-RBD IgG seropositivity after a single dose, with geometric mean (GM) optical density-450 (OD450) and sVNT inhibition of 1.96 and 81.3% on day 21 after dose 1 and 2.64 and 97.1% on day 28 after dose 2, respectively (Table 1). A high proportion (96.6%) of evaluable adolescent CC had positive S-RBD IgG after 2 doses, with GM OD450 value of 1.20 and GM sVNT inhibition of 71.2%. PRNT was performed for 60 BB and 64 CC age- and sex-matched adolescents, otherwise selected at random; GM for PRNT90 was 115 and 9.58 after BB and CC, respectively; GM for PRNT50 was 331 and 28.0 after BB and CC, respectively. In addition, these same 64 adolescent CC were also tested for N IgG and N-CTD IgG as secondary immunogenicity outcomes. N IgG and N-CTD IgG seropositivity in adolescent CC was high, at 98.4% and 92.2%, with GM OD450 of 1.72 and 2.09 across all responders and non-responders, while only 52.4% and 28.6% of 21 adult CC (GM OD450 of 0.77 and 0.92) selected at random were seropositive, respectively. S IgG, S IgG avidity and S IgG FcγRIIIa-binding were also performed, and the proportions of seropositivity were analogous to S-RBD IgG and sVNT (Table 1). After BB and CC, GM avidity indices were 29.7% and 20.5%, and the GM OD450 results of S IgG FcγRIIIa-binding, which is associated with antibody cellular cytotoxicity, were 2.07 and 0.75, respectively.

Compared to adults, humoral responses for the same vaccines were non-inferior for evaluable adolescent BB when measured by S IgG (GMR 1.09, 95% CI 1.03–1.15), S-RBD IgG (GMR 0.97, 95% CI 0.92–1.02), sVNT (GMR 1.02, 95% CI 1.02–1.03), PRNT90 (GMR 1.77, 95% CI 1.11–2.83), PRNT50 (GMR 1.28, 95% CI 0.84–1.96), S IgG avidity (GMR 1.26, 95% CI 1.15–1.38) and S IgG FcγRIIIa-binding (GMR 1.07, 95% CI 1.03–1.12) (Fig. 1b). Similarly, adolescents mounted non-inferior humoral

| | Vaccine regimen | Immunogenicity outcome | Geometric mean ratio plot | Result | GMR (95% CI) | P-value |
|---|---|---|---|---|---|---|
| **a** | BNT162b2, 1 dose | S IgG | | Inferior | 0.48 (0.42-0.54) | <0.0001 |
| | | S-RBD IgG | | Non-inferior and inferior | 0.72 (0.66-0.77) | <0.0001 |
| | | sVNT | | Non-inferior and inferior | 0.86 (0.84-0.88) | <0.0001 |
| | | PRNT90 | | Inferior | 0.22 (0.14-0.35) | <0.0001 |
| | | PRNT50 | | Inferior | 0.17 (0.10-0.30) | <0.0001 |
| | | S IgG avidity | | Non-inferior | 0.92 (0.82-1.03) | 0.13 |
| | | S IgG FcγRIIIa-binding | | Inferior | 0.58 (0.52-0.65) | <0.0001 |
| **b** | BNT162b2, 2 doses | S IgG | | Non-inferior and superior | 1.09 (1.03-1.15) | 0.0036 |
| | | S-RBD IgG | | Non-inferior | 0.97 (0.92-1.02) | 0.23 |
| | | sVNT | | Non-inferior and superior | 1.02 (1.02-1.03) | <0.0001 |
| | | PRNT90 | | Non-inferior and superior | 1.77 (1.11-2.83) | 0.018 |
| | | PRNT50 | | Non-inferior | 1.28 (0.84-1.96) | 0.25 |
| | | S IgG avidity | | Non-inferior and superior | 1.26 (1.15-1.38) | <0.0001 |
| | | S IgG FcγRIIIa-binding | | Non-inferior and superior | 1.07 (1.03-1.12) | 0.0005 |
| **c** | CoronaVac, 2 doses | S IgG | | Non-inferior and superior | 1.26 (1.07-1.48) | 0.0049 |
| | | S-RBD IgG | | Non-inferior | 1.00 (0.86-1.17) | 0.96 |
| | | sVNT | | Non-inferior and superior | 1.31 (1.15-1.48) | <0.0001 |
| | | PRNT90 | | Non-inferior | 1.24 (0.97-1.57) | 0.08 |
| | | PRNT50 | | Non-inferior | 1.30 (0.93-1.82) | 0.12 |
| | | S IgG avidity | | Non-inferior and superior | 1.72 (1.50-1.97) | <0.0001 |
| | | S IgG FcγRIIIa-binding | | Non-inferior | 1.25 (0.97-1.62) | 0.086 |
| | | N IgG | | Non-inferior and superior | 2.24 (1.87-2.68) | <0.0001 |
| | | N-CTD IgG | | Non-inferior and superior | 2.27 (1.82-2.82) | <0.0001 |

Geometric mean ratio plot x-axis: 0 — 0.6 (non-inferiority) — 1 — 2 — 3. Arrows: ← Adults better | Adolescents better →

**Fig. 1 Humoral immunogenicity outcomes for adolescents were mostly non-inferior in adolescents in comparison to adults. a** One dose of BNT162b2 (B) in adolescents was non-inferior by S-RBD IgG (adolescent B $N=107$, adult BB $N=115$), sVNT (adolescent B $N=107$, adult BB $N=115$) and S IgG avidity (adolescent B $N=88$, adult BB $N=114$) but not by S IgG (adolescent B $N=101$, adult BB $N=115$), PRNT90 (adolescent B $N=63$, adult BB $N=13$), PRNT50 (adolescent B $N=63$, adult BB $N=13$) and S IgG FcγRIIIa-binding (adolescent B $N=101$, adult BB $N=115$) (all $P<0.0001$, except S IgG avidity with $P=0.13$), which failed the non-inferiority comparison to adults. Additionally, although non-inferiority was satisfied for S-RBD IgG and sVNT for B in adolescents, their CIs were also within the inferior ranges (both $P<0.0001$). **b** In contrast, humoral responses in adolescents were non-inferior to adults after 2 doses of BNT162b2 (BB), as measured by S IgG (also superior, $P=0.0036$, adolescent BB $N=103$), S-RBD IgG ($P=0.23$, adolescent BB $N=103$), sVNT (also superior, $P<0.0001$, adolescent BB $N=104$), PRNT90 (also superior, $P=0.018$, adolescent BB $N=60$), PRNT50 ($P=0.25$, adolescent BB $N=60$), S IgG avidity (also superior, $P<0.0001$, adolescent BB $N=103$) and S IgG FcγRIIIa-binding (also superior, $P=0.0005$, adolescent BB $N=103$). **c** After 2 doses of CoronaVac (CC), adolescents also had non-inferior humoral responses to adults as assessed by S IgG (also superior, $P=0.0049$, adolescent CC $N=116$, adult CC $N=50$), S-RBD IgG ($P=0.96$, adolescent CC $N=119$, adult CC $N=51$), sVNT (also superior, $P<0.0001$, adolescent CC $N=119$, adult CC $N=51$), PRNT90 ($P=0.08$, adolescent CC $N=64$, adult CC $N=19$), PRNT50 ($P=0.12$, adolescent CC $N=64$, adult CC $N=19$), S IgG avidity (also superior, $P<0.0001$, adolescent CC $N=109$, adult CC $N=41$) and S IgG FcγRIIIa-binding ($P=0.086$, adolescent CC $N=116$, adult CC $N=50$). Additionally for adolescent CC, N and N-CTD IgGs were non-inferior and superior for adolescents compared to adults (both $P<0.0001$, adolescent CC $N=60$, adult CC $N=36$). GMR geometric mean ratio, CI confidence interval, S spike protein, RBD receptor-binding domain, N nucleocapsid protein, CTD C-terminal domain, sVNT surrogate virus neutralisation test, PRNT plaque reduction neutralisation test, FcγRIIIa Fcγ receptor IIIa.

responses after CC by S IgG (GMR 1.26, 95% CI 1.07–1.48), S-RBD IgG (GMR 1.00, 95% CI 0.86–1.17), sVNT (GMR 1.31, 95% CI 1.15–1.48), PRNT90 (GMR 1.24, 95% CI 0.97–1.57), PRNT50 (GMR 1.30, 95% CI 0.93–1.82), S IgG avidity (GMR 1.72, 95% CI 1.50–1.97) and S IgG FcγRIIIa-binding (GMR 1.25, 95% CI 0.97–1.62) (Fig. 1c). Interestingly, for N IgG and N-CTD IgG, only a small proportion of adult CC were seropositive (52.4% and 28.6% vs adolescent CC of 98.4% and 92.2%, respectively, $P < 0.0001$ for both). Adolescent CC satisfied the non-inferior and superior criterion (N IgG: GMR 2.24, 95% CI 1.87–2.68; N-CTD IgG: GMR 2.27, 95% CI 1.82–2.82) (Fig. 1c). N and N-CTD IgG levels were significantly elevated in adolescent CC (GM OD450 1.72 and 2.09, respectively) compared to adult CC (GM OD450 0.77 and 0.92, respectively), both $P < 0.0001$ (Table 1) (Supp. Fig. 2bii).

Adolescent B satisfied non-inferiority by S-RBD IgG (GMR 0.72, 95% CI 0.66–0.77), sVNT (GMR 0.86, 95% CI 0.84–0.88) and S IgG avidity (GMR 0.92, 95% CI 0.82–1.03), but not by S IgG (GMR 0.48, 95% 0.42–0.54), PRNT90 (GMR 0.22, 95% CI 0.14–0.35), PRNT50 (GMR 0.17, 95% CI 0.10–0.30) and S IgG FcγRIIIa-binding (GMR 0.58, 95% CI 0.52–0.65), which failed the non-inferiority criterion (Fig. 1a). Compared to adult BB, adolescent B satisfied the non-inferior and inferior criterion for S-RBD IgG and sVNT. Non-inferiority testing repeated in the expanded analysis populations confirmed similar findings (Supp. Table 2). S-RBD IgG, sVNT, PRNT90 and PRNT50 were all significantly lower in adolescent B than adult BB (GM OD450 1.96 vs 2.73, GM % inhibition 81.3% vs 94.9%, GM PRNT90 14.4 vs 64.6 and GM PRNT50 45.2 vs 259, respectively), all $P < 0.0001$ (Table 1) (Supp. Fig. 2a).

**Cellular immunogenicity outcomes between adolescents and adults.** Interferon-γ (IFN-γ)$^+$ and interleukin-2 (IL-2)$^+$ CD4$^+$ and CD8$^+$ T cells responses specific to S after B or BB or CC (and to N and M for CC) were analysed with intracellular cytokine staining on flow cytometry for 21–28 days post-dose 1 and 28 days post-dose 2 as primary outcomes (58 B, 56 BB and 60 CC evaluable adolescents were included; see Methods). A majority (≥70%) of adolescents receiving either vaccine had a detectable response (using ≥ 0.005% frequency of cytokine-expressing cells and stimulation index (SI) > 2, with DMSO used as the background negative control, as cut-off; see Methods) for S-specific IFN-γ$^+$CD4$^+$ or IL-2$^+$CD4$^+$ T cells 28 days after 2 doses, respectively (Table 2). In contrast, both IFN-γ$^+$CD8$^+$ and IL-2$^+$CD8$^+$ T cells specific to S were detectable in approximately half of adolescents receiving 2 doses of either vaccine. T cell responses to S after B and to SNM, N and M after CC are shown in Table 2.

S-specific IFN-γ$^+$CD4$^+$, IL-2$^+$CD4$^+$ and IFN-γ$^+$CD8$^+$ T cell responses satisfied the non-inferior criterion for evaluable adolescent BB compared to adults (GMR 1.23, 95% CI 0.66–2.29; GMR 1.15, 95% CI 0.67–1.99; GMR 1.32, 95% CI 0.64–2.73, respectively) (Fig. 2b). For adolescent CC compared to adults, SNM-specific (sum of responses to individual peptide pools) IL-2$^+$CD4$^+$, IFN-γ$^+$CD8$^+$ and IL-2$^+$CD8$^+$ (GMR 0.99, 95% CI 0.64–1.55; GMR 1.23, 95% CI 0.62–2.46; GMR 0.88, 95% CI 0.61–1.28, respectively), N-specific IFN-γ$^+$CD4$^+$, IL-2$^+$CD4$^+$, IFN-γ$^+$CD8$^+$ and IL-2$^+$CD8$^+$ (GMR 1.17, 95% CI 0.61–2.23; GMR 1.09, 95% CI 0.64–1.85; GMR 1.33, 95% CI 0.65–2.70; GMR 1.02, 95% CI 0.69–1.50, respectively), M-specific IFN-γ$^+$CD8$^+$ and IL-2$^+$CD8$^+$ T cells (GMR 1.25, 95% CI 0.66–2.35; GMR 0.95, 95% CI 0.66–1.37, respectively) were non-inferior (Fig. 2c). The remainder of the cellular immunogenicity outcomes between adolescents and adults were inconclusive as the 95% CI limits were out of the non-inferiority criterion of 0.60. All 4 S-specific T cell responses between adolescents and adults showed no

detectable differences in any vaccine regimens, except for S-specific IFN-γ$^+$CD4$^+$ T cell response, which was lower in adolescent B than adult BB (Supp. Fig. 3 and 3a, respectively). Similarly, the secondary analysis in the less stringent expanded analysis populations confirmed these findings of non-inferiority (Supp. Table 3).

**Immunogenicity assessments between BNT162b2 and CoronaVac in adolescents.** Antibody and T cell responses were compared between vaccines in evaluable adolescents at post-dose 1 and post-dose 2. CC elicited lower humoral responses than BB as measured by S IgG (GM OD450 0.54 vs 1.21; GMR 0.44, 95% CI 0.40–0.49), S-RBD IgG (GM OD450 1.20 vs 2.64; GMR 0.46, 95% CI 0.41–0.50), sVNT (GM % inhibition 71.2% vs 97.1%; GMR 0.73, 95% CI 0.68–0.79), PRNT90 (GM PRNT90 9.58 vs 115; GMR 0.08, 95% CI 0.07–0.11), PRNT50 (GM PRNT50 28.0 vs 331; GMR 0.08, 95% CI 0.07–0.11), S IgG avidity index (GM % avidity 20.5% vs 29.7%; GMR 0.69, 95% CI 0.63–0.76) and S IgG FcγRIIIa-binding (GM OD450 0.75 vs 2.07; GMR 0.36, 95% CI 0.31–0.42), all $P < 0.0001$ (Table 1a; Fig. 3a). Cellular immunogenicity outcomes were not significantly different except the S-specific IL-2$^+$CD4$^+$ T cell response was lower for CC (0.015% vs 0.032%; GMR 0.45, 95% CI 0.28–0.72) 28 days after 2 doses, $P = 0.001$ (Fig. 3b). Comparisons of N- and M- specific immunogenicity outcomes are not presented since CC but not BB induced non-spike responses, an expected finding.

Compared to their own baseline values, both evaluable BB and CC had significant increases in T cell responses for S-specific IFN-γ$^+$CD4$^+$ [BB GM fold rise (GMFR) 6.78, 95% CI 3.90–11.8; CC GMFR 3.99, 95% CI 2.35–6.78], IL-2$^+$CD4$^+$ (BB GMFR 7.20, 95% CI 4.84–10.7; CC GMFR 2.73, 95% CI 1.83–4.07) and IFN-γ$^+$CD8$^+$ T cells (BB GMFR 3.41, 95% CI 1.96–5.92; CC GMFR 3.49, 95% CI 1.98–6.15), respectively (all $P < 0.0001$) (Fig. 4a). As expected, there were no significant N-specific T cell responses elicited in BB. Additionally, CC elicited significant T cell responses and GMFR for S-specific IL-2$^+$CD8$^+$ (1.49, 95% CI 1.06-2.10, $P = 0.023$) (Fig. 4a), SNM-specific IFN-γ$^+$CD4$^+$ (3.58, 95% CI 2.35–5.44, $P < 0.0001$), IL-2$^+$CD4$^+$ (2.73, 95% CI 2.02–3.69, $P < 0.0001$) and IFN-γ$^+$CD8$^+$ (2.98, 95% CI 1.82–4.88, $P < 0.0001$) (Fig. 4b), N-specific IFN-γ$^+$CD4$^+$ (3.60, 95% CI 2.30–5.63, $P < 0.0001$), IL-2$^+$CD4$^+$ (3.95, 95% CI 2.71–5.75, $P < 0.0001$) and IFN-γ$^+$CD8$^+$ (1.81, 95% CI 1.02–3.21, $P = 0.042$) (Fig. 4c) and M-specific IL-2$^+$CD4$^+$ (1.45, 95% CI 1.06–2.00, $P = 0.022$) (Fig. 4d), respectively.

**Estimation of vaccine efficacies based on neutralisation titres for BNT162b2 and CoronaVac in adolescents.** Since neutralising antibodies have been established as a correlate of protection in multiple studies, we correlated our data with vaccine efficacies (VE) against symptomatic COVID-19 by using PRNT90 results normalised to convalescent sera in evaluable adolescents after receiving BNT162b2 or CoronaVac by mathematical extrapolation as previously published by Khoury et al (see Methods)[12,38,39]. One-dose BNT162b2 has been used as the primary series for adolescents in some localities, but not for adults or CoronaVac at any age, and therefore we included adolescent B, BB and CC only in this analysis. The mean neutralisation levels of BB, CC and B were 2.39, 0.20 and 0.30, which extrapolated to 93%, 50% and 59% VEs, respectively, all of which fulfilled the WHO's recommended 50% VE threshold as effective for use against COVID-19 (Fig. 5)[40]. This analysis was also repeated for PRNT50, which yielded similar findings.

**Reactogenicity and safety of BNT162b2 and CoronaVac in adolescents.** For adolescents in the healthy safety population,

**Table 2 Cellular immunogenicity outcomes in evaluable analysis populations by vaccine regimen.**

| | BNT162b2 | | | CoronaVac | |
|---|---|---|---|---|---|
| | Adolescents | Adolescents | Adults | Adolescents | Adults |
| | 1 dose | 2 doses | 2 doses | 2 doses | 2 doses |
| **T cell responses** | | | | | |
| **S-specific T cell responses on flow cytometry** | | | | | |
| N | 58 | 56 | 47 | 60 | 36 |
| GM % IFN-γ$^+$CD4$^+$ T cells | 0.014% | 0.041% | 0.033% | 0.023% | 0.021% |
| (95% CI) | (0.009-0.021%) | (0.028-0.06%) | (0.020-0.056%) | (0.015-0.036%) | (0.011-0.039%) |
| % positive (>/= cut-off at 0.005%) | 62.1%, $P = 0.21$ | 83.9%, $P = 0.33$ | 74.5% | 70.0%, $P = 0.38$ | 61.1% |
| GM % IL-2$^+$CD4$^+$ T cells | 0.023% | 0.032% | 0.028% | 0.015% | 0.015% |
| (95% CI) | (0.016-0.033%) | (0.023-0.045%) | (0.018-0.044%) | (0.011-0.020%) | (0.010-0.024%) |
| % positive (>/= cut-off at 0.005%) | 74.1%, $P = 0.82$ | 85.7%, $P = 0.31$ | 76.6% | 73.3%, $P = 0.82$ | 69.4% |
| GM % IFN-γ$^+$CD8$^+$ T cells | 0.009% | 0.018% | 0.013% | 0.014% | 0.015% |
| (95% CI) | (0.006-0.014%) | (0.011-0.028%) | (0.008-0.024%) | (0.009-0.023%) | (0.007-0.029%) |
| % positive (>/= cut-off at 0.005%) | 41.4%, $P = 0.69$ | 57.1%, $P = 0.33$ | 46.8% | 48.3%, $P = 0.83$ | 44.4% |
| GM % IL-2$^+$CD8$^+$ T cells | 0.005% | 0.005% | 0.007% | 0.006% | 0.007% |
| (95% CI) | (0.004-0.007%) | (0.004-0.007%) | (0.005-0.010%) | (0.005-0.008%) | (0.004-0.010%) |
| % positive (>/= cut-off at 0.005%) | 32.8%, $P = 0.11$ | 44.6%, $P = 0.70$ | 48.9% | 48.3%, $P = 0.67$ | 41.7% |
| **Total S, N and M-specific T cell responses on flow cytometry** | | | | | |
| N | / | / | / | 60 | 36 |
| GM % IFN-γ$^+$CD4$^+$ T cells | / | / | / | 0.058% | 0.068% |
| (95% CI) | | | | (0.041-0.083%) | (0.041-0.113%) |
| % positive (>/= cut-off at 0.01%) | / | / | / | 83.3%, $P = 0.59$ | 77.8% |
| GM % IL-2$^+$CD4$^+$ T cells | / | / | / | 0.039% | 0.040% |
| (95% CI) | | | | (0.030-0.052%) | (0.027-0.057%) |
| % positive (>/= cut-off at 0.01%) | / | / | / | 83.3%, $P = 0.59$ | 77.8% |
| GM % IFN-γ$^+$CD8$^+$ T cells | / | / | / | 0.050% | 0.041% |
| (95% CI) | | | | (0.033-0.077%) | (0.023-0.071%) |
| % positive (>/= cut-off at 0.01%) | / | / | / | 65.0%, $P = 0.52$ | 58.3% |
| GM % IL-2$^+$CD8$^+$ T cells | / | / | / | 0.017% | 0.020% |
| (95% CI) | | | | (0.014-0.022%) | (0.014-0.027%) |
| % positive (>/= cut-off at 0.01%) | / | / | / | 58.3%, $P > 0.9999$ | 58.3% |
| **N-specific T cell responses on flow cytometry** | | | | | |
| N | / | / | / | 60 | 36 |
| GM % IFN-γ$^+$CD4$^+$ T cells | / | / | / | 0.011% | 0.010% |
| (95% CI) | | | | (0.008-0.017%) | (0.006-0.016%) |
| % positive (>/= cut-off at 0.005%) | / | / | / | 55.0%, $P = 0.68$ | 50.0% |
| GM % IL-2$^+$CD4$^+$ T cells | / | / | / | 0.013% | 0.012% |
| (95% CI) | | | | (0.009-0.018%) | (0.008-0.018%) |
| % positive (>/= cut-off at 0.005%) | / | / | / | 66.7%, $P > 0.9999$ | 66.7% |
| GM % IFN-γ$^+$CD8$^+$ T cells | / | / | / | 0.008% | 0.006% |
| (95% CI) | | | | (0.005-0.012%) | (0.003-0.010%) |
| % positive (>/= cut-off at 0.005%) | / | / | / | 31.7%, $P = 0.64$ | 25.0% |
| GM % IL-2$^+$CD8$^+$ T cells | / | / | / | 0.004% | 0.004% |
| (95% CI) | | | | (0.003-0.005%) | (0.003-0.006%) |
| % positive (>/= cut-off at 0.005%) | / | / | / | 28.3%, $P = 0.81$ | 25.0% |
| **M-specific T cell responses on flow cytometry** | | | | | |
| N | / | / | / | 60 | 36 |
| GM % IFN-γ$^+$CD4$^+$ T cells | / | / | / | 0.007% | 0.009% |
| (95% CI) | | | | (0.005-0.010%) | (0.005-0.016%) |
| % positive (>/= cut-off at 0.005%) | / | / | / | 36.7%, $P = 0.67$ | 41.7% |
| GM % IL-2$^+$CD4$^+$ T cells | / | / | / | 0.006% | 0.006% |
| (95% CI) | | | | (0.004-0.007%) | (0.004-0.009%) |
| % positive (>/= cut-off at 0.005%) | / | / | / | 46.7%, $P > 0.9999$ | 47.2% |
| GM % IFN-γ$^+$CD8$^+$ T cells | / | / | / | 0.006% | 0.005% |
| (95% CI) | | | | (0.004-0.009%) | (0.003-0.008%) |
| % positive (>/= cut-off at 0.005%) | / | / | / | 25.0%, $P = 0.62$ | 19.4% |
| GM % IL-2$^+$CD8$^+$ T cells | / | / | / | 0.004% | 0.004% |
| (95% CI) | | | | (0.003-0.005%) | (0.003-0.006%) |
| % positive (>/= cut-off at 0.005%) | / | / | / | 23.3%, $P > 0.9999$ | 25.0% |

*GM* geometric mean, *CI* confidence interval, *IFN-γ* interferon-gamma, *IL-2* interleukin-2.
*P*-values compare the proportion of positive responses between adolescents receiving 1 or 2 doses of vaccine and adults receiving 2 doses of the same vaccine by two-tailed Fisher exact test.

pain at the injection site was the most common AR reported for both vaccines, which was significantly more for those who received BNT162b2 ($N = 116$) than CoronaVac ($N = 123$) (B: 89.7% vs C: 54.5%, $P < 0.0001$; BB: 87.9% vs CC: 52.9%, $P < 0.0001$) (Fig. 6). BNT162b2 was also associated with more reports of several other ARs. More participants had antipyretics use after either dose of BNT162b2 (B: 9.5% vs C: 1.6%, $P = 0.009$; BB: 22.4% vs CC: 0.8%, $P < 0.0001$). Twenty-six mild AEs and 4 moderate AEs were reported within 28 days after vaccination in adolescents (22 for BB and 8 for CC in total) (Supp. Table 4). There was no serious adverse event (SAE) for either vaccine.

**Correlation of assays, characteristics and haematological parameters in adolescents**. As a secondary objective, we explored potential associations between the immunogenicity outcomes (see Methods) in evaluable adolescents after B (Supp. Fig. 4a) and C (Supp. Fig. 4b). Immunogenicity outcomes after 2 doses of vaccines were high for many participants, and therefore the correlation analyses were performed for post-dose 1 only. There was strong correlation within humoral (S IgG with S IgG FcγRIIIa-binding, and sVNT with PRNT50) and within cellular outcomes (S-specific IFN-γ$^+$CD4$^+$ with IL-2$^+$CD4$^+$ and IFN-γ$^+$CD8$^+$; IL-2$^+$CD8$^+$ with IFN-γ$^+$CD8$^+$) for both vaccines. There was no

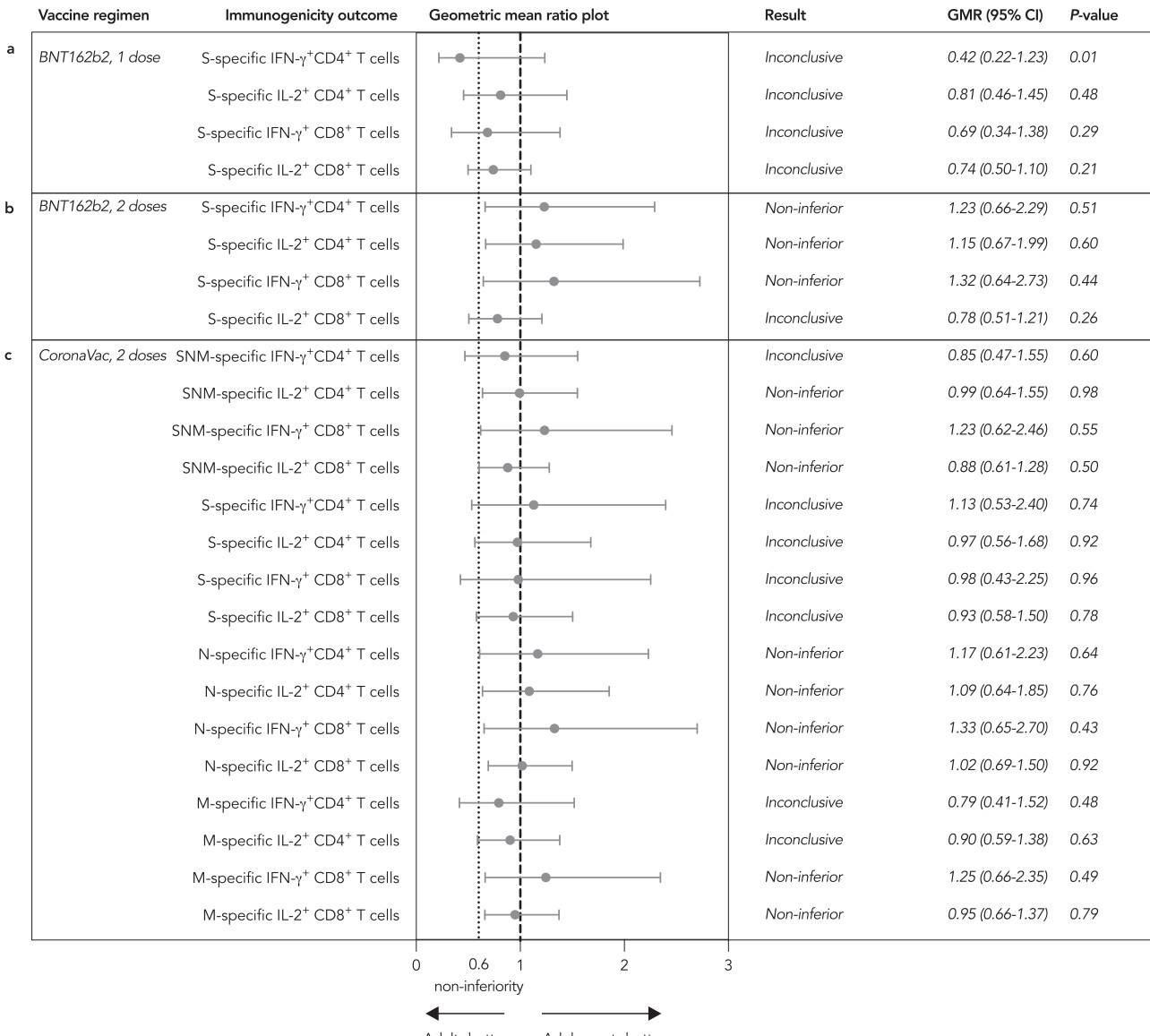

**Fig. 2 Cellular immunogenicity outcomes for adolescents were mostly non-inferior or inconclusive in adolescents in comparison to adults.**
**a**, **b** Adolescents receiving one dose of BNT162b2 (B; $N = 58$) and 2 doses of BNT162b2 (BB; $N = 56$) and **c** adolescents receiving 2 doses of CoronaVac (CC; $N = 60$) were tested for IFN-$\gamma^+$ and IL-2$^+$ CD4$^+$ and CD8$^+$ T cells on flow-cytometry-based intracellular cytokine staining assays specific to S (and N and M for CC) for 21 days after dose 1 and 28 days after dose 2. The results of SNM-specific T cell responses were calculated from the sum of responses of the individual S, N and M peptide pools. S-specific IFN-$\gamma$+CD4+, IL-2+CD4+ and IFN-$\gamma$+CD8$^+$ T cell responses were non-inferior for adolescent BB in comparison to adults ($N = 47$). For adolescent CC compared to adults ($N = 36$), SNM-specific IL-2+CD4+, IFN-$\gamma$+CD8$^+$ and IL-2+CD8$^+$, N-specific IFN-$\gamma$+CD4+, IL-2+CD4+, IFN-$\gamma$+CD8$^+$ and IL-2+CD8$^+$, M-specific IFN-$\gamma$+CD8$^+$ and IL-2+CD8$^+$ were non-inferior. The remaining cellular immunogenicity outcomes were inconclusive. Dots and error bars show GMR estimates and two-sided 95% CI respectively. GMR geometric mean ratio, CI confidence interval, S spike protein, N nucleocapsid protein, M membrane protein, IFN-$\gamma$ interferon-$\gamma$, IL-2 interleukin-2.

correlation between the humoral and cellular outcomes. We also explored associations between sVNT levels and baseline demographic, anthropometric (including weight for height for age and sex) and haematological variables (total white blood cell count, absolute lymphocyte count, haemoglobin concentration) after B or C, which yielded no significant findings (Supp. Tables 5 and 6). The participants were generally healthy without major associated co-morbidities.

## Discussion
This study assessed the immunogenicity profiles of 2 major platforms of vaccines against COVID-19 in adolescents. Overall, the data demonstrated that most antibody and T cell responses for 2 doses of mRNA-based BNT162b2 and inactivated CoronaVac vaccines in 11- to 17-year-old children were non-inferior compared to adults. Between vaccines, the antibody levels were higher for adolescent BB than CC. S-specific T cell responses after 2 doses were robust and similar in adolescents receiving either vaccine, and N- and M-specific T cells were detected after CoronaVac but not BNT162b2, due to the absence of these antigens in BNT162b2. There was no or low correlations between antibody and T cell responses at the studied timepoint, and the reasons for this are unclear. One possibility is that the link between the two arms of the adaptive immune system is dynamic, and the observed discordance was possibly related to response kinetics

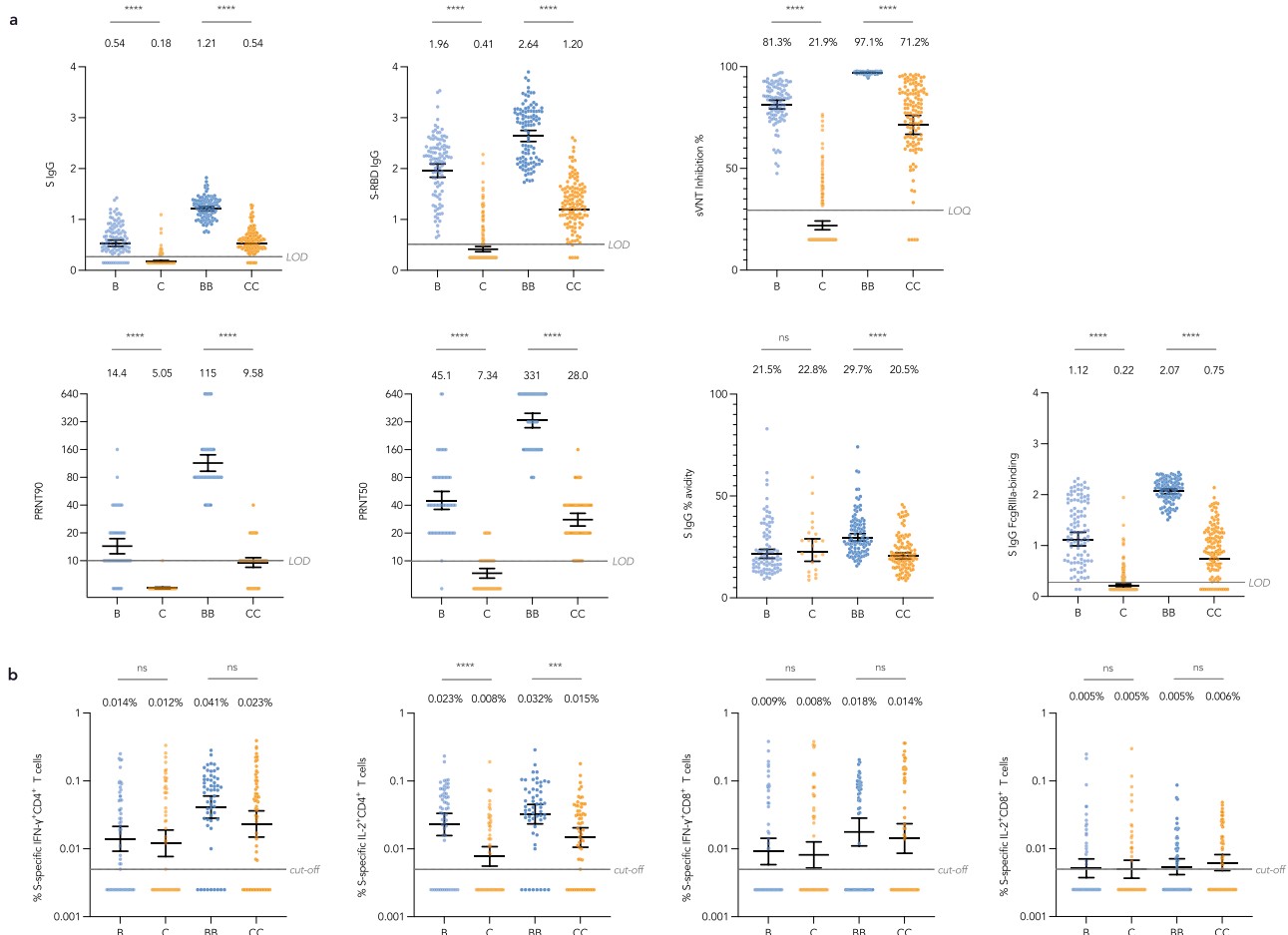

**Fig. 3 Antibody levels against S were higher for BNT162b2 than CoronaVac in adolescents.** Humoral and cellular immunogenicity was compared between vaccines in adolescents at 21-28 days after 1 dose and 28 days after 2 doses. **a** There were lower humoral responses after CC than BB as measured by S IgG (adolescent CC $N = 116$, adolescent BB $N = 103$) (GM OD450 0.54 vs 1.21; GMR 0.44, 95% CI 0.40–0.49), S-RBD IgG (adolescent CC $N = 119$, adolescent BB $N = 104$) (GM OD450 1.20 vs 2.64; GMR 0.46, 95% CI 0.41–0.50), sVNT (adolescent CC $N = 119$, adolescent BB $N = 104$) (GM % inhibition 71.2% vs 97.1%; GMR 0.73, 95% CI 0.68-0.79), PRNT90 (adolescent CC $N = 64$, adolescent BB $N = 60$) (GM PRNT90 9.58 vs 115; GMR 0.08, 95% CI 0.07–0.11), PRNT50 (adolescent CC $N = 64$, adolescent BB $N = 60$) (GM PRNT50 28.0 vs 331; GMR 0.08, 95% CI 0.07–0.11), S IgG avidity index (adolescent CC $N = 109$, adolescent BB $N = 103$) (GM % avidity 20.5% vs 29.7%; GMR 0.69, 95% CI 0.63–0.76) and S IgG FcγRIIIa-binding (adolescent CC $N = 116$, adolescent BB $N = 103$) (GM OD450 0.75 vs 2.07; GMR 0.36, 95% CI 0.31–0.42) (all $P < 0.0001$). Most outcomes except S IgG avidity were also lower in C compared to B. **b** Cellular immunogenicity outcomes were similar between vaccine types except for the S-specific IL-2⁺CD4⁺ T cell response (adolescent CC $N = 60$, adolescent BB $N = 56$), which was lower after CC (GM % T cells 0.015% vs 0.032%; GMR 0.45, 95% CI 0.28-0.72) ($P = 0.001$). Nucleocapsid (N) and membrane (M)-specific T cell responses were not compared since only CC but not BB had induced non-spike responses, as expected. Data labels and centre lines show GM estimates, and error bars show 95% CI. $P$-values were derived from two-tailed unpaired t test after natural logarithmic transformation. GM geometric mean, GMR geometric mean ratio, CI confidence interval, B 1 dose of BNT162b2, BB 2 doses of BNT162b2, C 1 dose of CoronaVac, CC 2 doses of CoronaVac, S spike protein, RBD receptor-binding domain, sVNT surrogate virus neutralisation test, PRNT plaque reduction neutralisation test, FcγRIIIa Fcγ receptor IIIa, IFN-γ interferon-γ, IL-2 interleukin-2. ***$P < 0.001$; ****$P < 0.0001$.

not being observed at the sampling timeframe. Both vaccines were associated with transient, tolerable AR in this age group, which occurred more frequently for BNT162b2 than CoronaVac. We investigated T cell responses to COVID-19 vaccines in adolescents and found that adolescents mounted far superior antibody responses against N compared to adults.

Approval of vaccines in the younger age group to prevent hospitalisations and reduce COVID-19 severity is key during and after the transition of the devastating pandemic to benign endemicity, which requires infections in all ages including children to be mild[41]. The pivotal phase 3 trial for BNT162b2 in adolescents tested for non-inferiority of neutralising antibody titres to support extending authorisation of use to this age group and found superior PRNT in adolescents versus young adults[9]. While our study was not powered to determine superiority, our finding of

non-inferiority was consistent. The phase 2 licensing trial on CoronaVac did not formally test for non-inferiority; however, we also showed that PRNT GM titres were non-inferior, supporting the use of CoronaVac in adolescents in the absence of efficacy data. Due to a high incidence of myocarditis after 2 doses of BNT162b2 in young adults, especially male adolescents, the second dose for adolescents was held off in places such as HK and the UK when there was no effectiveness data to inform this policy[34,35]. It was therefore relevant to ask if a single dose of vaccine could be non-inferior to the adult 2-dose schedule since adolescents mounted better antibody responses to 2 doses of vaccine. However, we found an inferior neutralising antibody response in adolescents receiving a single dose of BNT162b2 to adults who received 2 doses. This implies that a single dose may not be as effective in reducing symptomatic COVID-19 and other

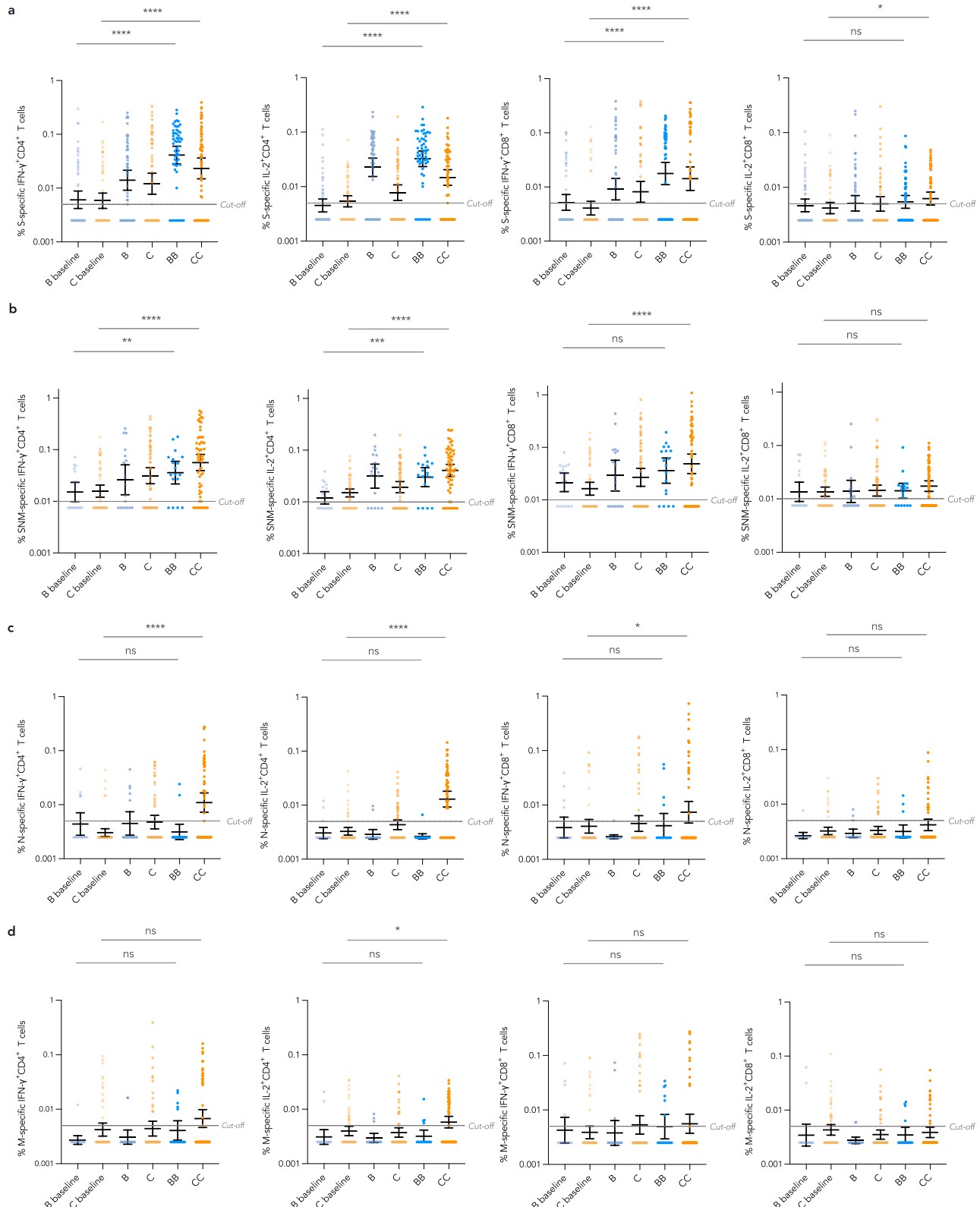

measures to mitigate the risk of myocarditis, such as increased dosing interval, should be used[42].

We demonstrated similar or non-inferior T cell response in adolescent vaccinees. While a COP has yet to be confirmed for protection against severe COVID-19, CD8⁺ T cells are likely a key defence against disease progression[16,43]. The concept that T cell immunity can be important for controlling severe viral infections is not novel. Burnet proposed in 1968 that humoral immunity did not mediate the "eruptive stage of measles" as "measles follows its normal course" in patients with agammaglobulinemia[44]. This is confirmed for COVID-19 in patients with severe combined immunodeficiencies and have absent or a malfunctional T cell compartment, who showed a high rate of fatality, which was not observed in those with

**Fig. 4 Significant increases in N- and M-specific T cell responses after CoronaVac in adolescents. a** For adolescents who received each vaccine, when compared to their own baseline values, BB (N = 56) and CC (N = 60) had significant increases in T cell responses for S-specific IFN-γ⁺CD4⁺, IL-2⁺CD4⁺ and IFN-γ⁺CD8⁺ (all P < 0.0001). Additionally, a significant increase in S-specific IL-2⁺CD8⁺ T cells was observed for CC (P = 0.023). **b** When added together, SNM-specific IFN-γ⁺CD4⁺ (P < 0.0001), IL-2⁺CD4⁺ (P < 0.0001) and IFN-γ⁺CD8⁺ T cells (P < 0.0001) increased significantly for CC (N = 60). **c** These marked increases were likely due to post-CC's combined increases in S-specific T cell responses as well as N-specific increases in IFN-γ⁺CD4⁺ (P < 0.0001), IL-2⁺CD4⁺ (P < 0.0001), IFN-γ⁺CD8⁺ (P = 0.042) and **d** M-specific IL-2⁺CD4⁺ (P = 0.021). On the other hand, no significant N- and M-specific T cell responses were elicited by BB, an expected result. Centre lines show GM estimates, and error bars show 95% CI. P-values were derived from two-tailed paired t test after natural logarithmic transformation. GM geometric mean, CI confidence interval, B 1 dose of BNT162b2, BB 2 doses of BNT162b2, C 1 dose of CoronaVac, CC 2 doses of CoronaVac, S spike protein, N nucleocapsid protein, M membrane protein, IFN-γ interferon-γ, IL-2 interleukin-2. *P < 0.05; **P < 0.01; ***P < 0.001; ****P < 0.0001.

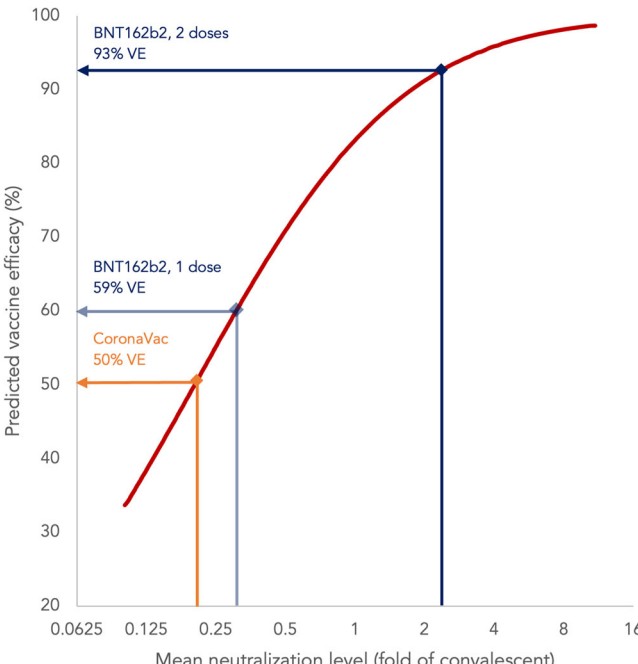

**Fig. 5 Vaccine efficacy estimates based on neutralising antibody titres for BNT162b2 (after 1 dose or 2 doses) and CoronaVac (after 2 doses) were ≥50% in adolescents.** Neutralising antibodies have been established as a reliable correlate of protection that can predict VEs against symptomatic COVID-19. The mean neutralising levels (fold of convalescent) were derived by dividing the geometric mean titres of PRNT90 in healthy evaluable adolescents who received the vaccines with that of 102 convalescent sera collected on days 28–59 post-onset of illness in patients aged ≥18 years. A point estimate of VE was extrapolated from the best fit of the logistic model in Khoury et al.[12,38,39]. Adolescent B has been considered completion of primary series, but not adolescent CC or adult B, for a time period in HK and the UK due to elevated myocarditis risks after youths received 2 doses of BNT162b2. Therefore, the VE of adolescent B, but not adolescent C or adult CC, was also extrapolated, along with adolescent BB and CC. The mean neutralisation levels (fold of convalescent) for adolescents after receiving 2 doses of BNT162b2, 2 doses of CoronaVac and 1 dose of BNT162b2 were 2.39, 0.20 and 0.30, respectively. Extrapolation of these mean neutralisation levels using the logistic model resulted in VEs of 93% after 2 doses of BNT162b2, 50% after 2 doses of CoronaVac and 59% after 1 dose of BNT162b2. VE, vaccine efficacy.

X-linked agammaglobulinemia[26]. In hepatitis B immunisation, vaccine effectiveness remains potent many years after vaccination with long-lived T and B cell responses and vastly waned antibody titres[45,46]. Moreover, despite exponential differences in neutralising antibody titres, SARS-CoV-2 vaccines of different platforms including mRNA, adenoviral vector and inactivated vaccines have been shown to produce potent T cell responses and very high effectiveness against hospitalisation[31,47–49]. Multiple lines of evidence support the contention that T cells play a major role in mediating protection against severe COVID-19[31,49–56]. Therefore, our T cell results suggest adolescents receiving either vaccine are also protected from severe COVID-19.

There were few studies that investigated immunogenicity between the mRNA and inactivated vaccines by direct comparison. In adults, BNT162b2 induced the strongest neutralising antibody response on sVNT, followed by the adenovirus viral vector vaccines ChAdOx1/nCoV-19 and then Gam-COVID-Vac, and lastly the inactivated BBIBP-CorV. This pattern of antibody response was similar across many other variants of concern tested[57]. Another study in adults involving our group also found BNT162b2 elicited higher neutralising antibody titres, antibody Fc receptor binding and antibody avidity than CoronaVac, while 20 of 49 (40.8%) from the CoronaVac group had N-CTD IgG on ELISA[31]. In line with the stronger S IgG responses, our study observed that a similar 6/21 (28.6%) of adults, but a majority (59/64, 92.2%) of adolescents, developed IgG against N-CTD. N-CTD is responsible for type I interferon antagonism of the N protein[22]. Additionally, inhibitors of GSK-3 that activates N impair SARS-CoV-2 replication in lung epithelial cells[58]. The clinical relevance of N is illustrated by the observation that patients taking lithium, which has inhibitory activity against GSK-3, have reduced risk of COVID-19[58]. However, the underlying mechanisms and implications of the higher N and N-CTD IgGs differentially in adolescents in this study are unknown and deserve further research. CoronaVac appeared to elicit greater CD4⁺ and CD8⁺ T cell responses against the SARS-CoV2 structural peptide pool than BNT162b2 in a small group of adults[31], and we also observed N- and M-specific T cell responses in adolescents receiving CC.

Recipients of CoronaVac appear to mount lower antibody titres than BNT162b2. Our group and others had previously shown that intradermal administration of inactivated influenza vaccine can enhance humoral responses across different age groups[59–61]. Whether this vaccination route induces higher humoral responses against SARS-CoV-2 and its VOCs should be explored for CoronaVac, which is especially important for countries where mRNA vaccines are not available.

In this study, every participant was evaluated and followed by physicians and nurses, supported by an online AR reporting system. Despite its reliable approach, there were several limitations, such as the unblinded, non-randomised design. Our intended objective was to assess the results of a real-life, practical approach. Because of the non-randomised design, there is potential for selection bias. However, age and sex were similar between participants receiving both vaccines and we expect that the comparisons of immunogenicity are valid. Thirty-three adults received CC but did not have blood sampling within the evaluable interval, since CoronaVac has been available to adults in Hong Kong for 13-14 weeks prior to study initiation. However, non-inferiority testing in the evaluable and expanded analysis

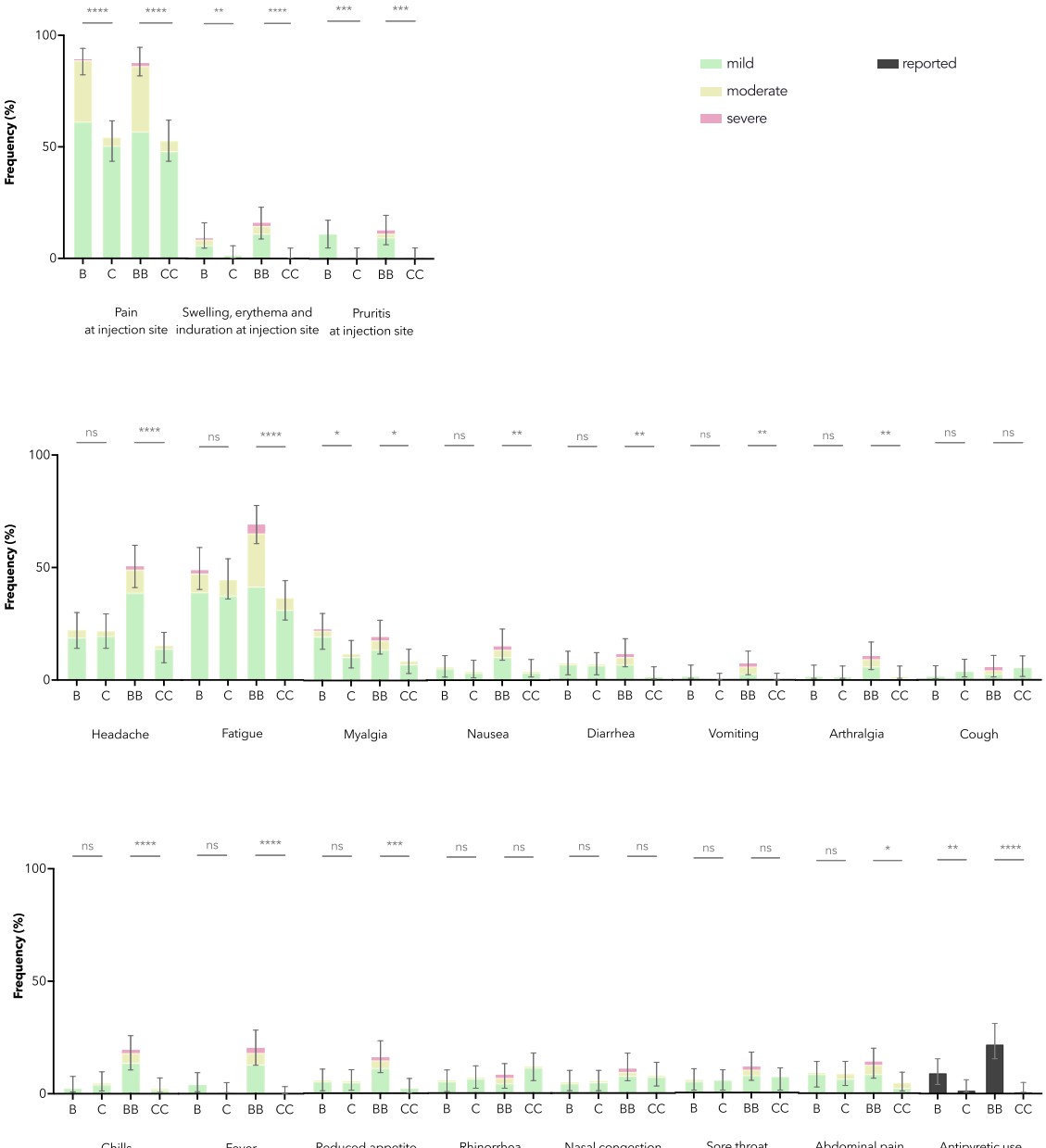

**Fig. 6 Adverse reactions 7 days after each dose of BNT162b2 and CoronaVac were solicited from adolescents in the healthy safety population.** In the adolescent healthy safety population, pain at the injection site was the most common adverse reaction (ARs) reported for both vaccines, which was significantly more for those who received BNT162b2 ($N = 116$) than CoronaVac ($N = 123$) (B: 89.7% vs C: 54.5%, $P < 0.0001$; BB: 87.9% vs CC: 52.9%, $P < 0.0001$). BNT162b2 was also associated with more reporting of several other ARs, including swelling, erythema, induration and pruritis at the injection site, headache, fatigue, myalgia, nausea, diarrhoea, vomiting, arthralgia, chills, fever, reduced appetite, and abdominal pain. More participants had antipyretics use after either dose of BNT162b2 than CoronaVac (B: 9.5% vs C: 1.6%, $P = 0.009$; BB: 22.4% vs CC: 0.8%, $P < 0.0001$). Data are shown as percentages and error bars show two-sided 95% CI of the total frequency of the respective AR of any severity. CI confidence interval, B 1 dose of BNT162b2, BB 2 doses of BNT162b2, C 1 dose of CoronaVac, CC 2 doses of CoronaVac. *$P < 0.05$; **$P < 0.01$; ***$P < 0.001$; ****$P < 0.0001$.

populations yielded consistent findings, ensuring the validity of our results. Transmission of COVID-19 has been aggressively contained by the HK Government, with close contact tracing and quarantine measures, which precluded differentiation of the clinical efficacies of the 2 vaccines. However, immunobridging data are important, and we were able to estimate clinical efficacy based on neutralising antibodies COP. It is necessary to note that estimating VEs based on immune response curves, though an important advancement during the pandemic, is still no replacement for epidemiologically rigorous VE studies. Finally, this

study did not assess immunogenicity against variants of concern (VOC), such as the newly emerged B.1.1.529 (Omicron) variant, which will be an important future step.

There is a major concern that Omicron's multiple S mutations may allow its escape from immunity after natural infection or vaccination. Multiple reports suggested 2 doses of BNT162b2 and CoronaVac failed to generate Omicron-neutralising antibodies in a majority of adults[62]. A BNT162b2 booster elicits neutralising antibodies in most adults, although their durability may be limited[63]. In contrast, several studies found that T cell responses

against the Omicron S protein are largely preserved (>80%) in most vaccinated and previously infected adults[49,51–55]. This lasting cellular immunity is one of many possibilities that contributes to the high clinical effectiveness of 2 doses of BNT162b2 doses against hospitalisation and death during the initial Omicron wave[64–68]. Since this study showed that vaccine-induced T cell responses in adolescents were similar to adults, it is likely that their T cell responses remain protective against VOCs, including Omicron. We speculate that immunisation platforms, such as the inactivated vaccines or multivalent peptide vaccines, which contain more conserved coronavirus protein antigens other than S, could be less susceptible to T cell escape and reduced clinical effectiveness against hospitalisation[56]. Taken together, vaccination elicits robust immune responses and remains a key method for providing host protection against COVID-19 in adolescents.

## Methods

**Study design.** Coronavirus disease-19 (COVID-19) Vaccination in Adolescents and Children (COVAC) is a registered clinical study (Department of Health, Hong Kong (HK), Clinical Trial Certificate 101894; clinicaltrials.gov NCT04800133) with a non-inferiority, non-blinded, non-randomised design aimed at establishing immunobridging for 2 COVID-19 vaccines, BNT162b2 (B) and CoronaVac (C), in children. Additionally, the study aimed to compare the reactogenicity and immunogenicity between the 2 vaccines in children. The research protocol and procedures were approved by the University of Hong Kong (HKU)/HK West Cluster Hospital Authority Institutional Review Board (UW21-157) and in compliance with the October 2013 Declaration of Helsinki principles, which were performed at a community vaccination centre (CVC) supported by HKU under the government's COVID-19 immunisation programme.

**Participants.** This pre-specified interim analysis included 11- to 17-year-old children and ≥18-year-old adults who were family members and unrelated individuals. Recruitment targeted schoolchildren across HK. Potential participants needed to be healthy or in stable health condition, and those with known history of COVID-19 positivity at any of the 3 study visits (also baseline S-RBD IgG positivity, or ORF8 IgG positivity at any visit, or N IgG positivity at any visit for B or BB), history of severe allergy, significant neuropsychiatric conditions, immunocompromised states, transfusion of blood products within 60 days, haemophilia, pregnancy or breastfeeding were excluded from this analysis.

**Procedures.** Potential participants were recruited via schools, mass media or referral. Study physicians provided information to participants and their parents/ legally acceptable representatives (LARs), obtained informed consent from participants aged 18 years or above, or for underage participants, from their parents and LARs. Assent was also obtained from underage participants. Peripheral blood was then taken before each dose, 4 weeks after the second-dose B (BB) and second-dose C (CC). The 2 doses of B and C were given 21-28 and 28-35 days apart, respectively. The dosages of BNT162b2 and CoronaVac were 0.3 mL (equivalent to 30 µg of COVID-19 mRNA vaccine embedded in lipid nanoparticles) and 0.5 mL (600 SU, equivalent to 3 µg, of inactivated SARS-CoV-2 virus as antigen), respectively.

*Safety data collection.* Participants were observed by study nurse(s) for 30 min after receiving the vaccine and attended by study physician(s) if clinically indicated. The study protocol required their recording of no or any prespecified adverse reactions (ARs) in an online or paper-based diary for 7 days. Adverse events (AEs) not included in the list were to be manually reported up to 28 days. Severe adverse events (SAEs), i.e., hospitalisations, life-threatening complications, disabilities, deaths and birth defects of their offspring, breakthrough COVID-19, would be followed for 3 years. These cases were reviewed by the study physicians, who determined the possibility of clinical relevance to the study vaccine.

*S-RBD IgG, N and N-CTD IgG, surrogate virus neutralisation assay (sVNT) and plaque reduction neutralisation test (PRNT).* Peripheral clotted blood was drawn, and the serum was stored at −80 °C after separation. The SARS-CoV-2 S receptor-binding domain (S-RBD) IgG enzyme-linked immunosorbent assay (ELISA) and PRNT were carried out as described below, which has been validated in our previous publication[69]. sVNT was conducted according to the manufacturer's instructions (GenScript Inc, Piscataway, USA) as described below, which has been previously validated[38,69]. All sera were heat-inactivated at 56 °C for 30 min before testing.

In brief, S-RBD IgG ELISA plates were coated overnight with 100 ng/well of purified recombinant S-RBD in PBS buffer, followed by addition of 100 µL Chonblock Blocking/Sample Dilution (CBSD) ELISA buffer (Chondrex Inc, Redmond, USA). This was incubated at room temperature (RT) for 2 hours. Serum was tested at a dilution of 1:100 in CBSD ELISA buffer, then added to the wells for

2 hours at 37 °C. After washing with PBS containing 0.1% Tween 20, horseradish peroxidase (HRP)-conjugated goat anti-human IgG (1:5000, Thermo Fisher Scientific #31410) was added for 1 hour at 37 °C, followed by washing five times with PBS containing 0.1% Tween 20. HRP substrate (Ncm TMB One, New Cell & Molecular Biotech Co. Ltd, China) of 100 µL was added for 15 min, and the reaction was stopped by 50 µL of 2 M H2SO4. The OD was analysed in a Sunrise absorbance microplate reader (Tecan, Männedorf, Switzerland) at 450 nm wavelength. The background OD in PBS-coated control wells with the participant's serum was subtracted from each OD reading. Values at or above an OD450 of 0.5 were considered positive and values below were imputed as 0.25.

For N and N-CTD IgG, 96-well ELISA plates (Nunc MaxiSorp, Thermo Fisher Scientific) were first coated overnight with 125 ng (full length N) or 40.3 ng (N-CTD) per well of purified recombinant protein in PBS buffer. The plates were then blocked with 100 µl of Chonblock blocking/sample dilution ELISA buffer (Chondrex Inc, Redmon, US), followed by incubation at room temperature for 1 h. Each human plasma sample was diluted to 1:100 in Chonblock blocking/sample dilution ELISA buffer. Each sample was then added into the ELISA plates for a two-hour incubation at 37 °C. After extensive washing with PBS containing 0.1% Tween 20, each well in the plate was further incubated with the anti-human IgG secondary antibody (1:2500, Thermo Fisher Scientific #31410) for 1 hour at 37 °C. The ELISA plates were then washed five times with PBS containing 0.1% Tween 20. Subsequently, 100 µL of HRP substrate (Ncm TMB One; New Cell and Molecular Biotech Co. Ltd, Suzhou, China) was added into each well. After 15 min of incubation, the reaction was stopped by adding 50 µL of 2 M H2SO4 solution and analysed on an absorbance microplate reader at 450 nm wavelength.

The sVNT was performed using 10 µL of each serum, positive and negative controls, which were diluted at 1:10 and mixed with an equal volume HRP conjugated to the wild-type SARS-CoV-2 S-RBD) (6 ng). The mixture was incubated for 30 min at 37 °C, then 100 µL of each sample was added to microtitre plate wells coated with angiotensin-converting enzyme-2 (ACE-2) receptor. This plate was sealed for 15 min at 37 °C and then washed with wash-solution, tapped dry, and 100 µL of 3,3',5,5'-tetramethylbenzidine (TMB) was added and incubated in the dark at RT for 15 min. This reaction was terminated using 50 µL of Stop Solution and the absorbance was read at 450 nm in a microplate reader. After confirming the positive and negative controls provided the recommended OD450 values, the % inhibition of each serum was calculated as (1 - sample OD value/ negative control OD value) x100%. Inhibition (%) of at least 30%, the limit of quantification (LOQ), was regarded as positive, and values below 30% were imputed as 10%.

The PRNT was performed in duplicate using culture plates (Techno Plastic Products AG, Trasadingen, Switzerland) in a biosafety level 3 facility[38,39,69]. Serial serum dilutions from 1:10 to at least 1:320 were incubated with ~300 plaque-forming units of the wild-type strain, SARS-CoV-2 BetaCoV/Hong Kong/ VM20001061/2020 virus, for 1 hour at 37 °C. The virus-serum mixtures were added on to Vero-E6 cell monolayers and incubated for 1 hour at 37 °C in a 5% CO2 incubator. The plates were overlaid with 1% agarose in cell culture medium and incubated for 3 days when the plates were fixed and stained. Antibody titres were defined as the reciprocal of the highest serum dilution that resulted in >90% (PRNT90, a more stringent cut-off) or >50% (PRNT50) reduction in the number of plaques. Values below the lowest dilution tested (10) were imputed as 5 and those above 320 were imputed as 640.

*S IgG, avidity and FcγRIIIa-binding.* Detection of S IgG, avidity and FcγRIIIa-binding was carried out with reference to previous experiments[31]. Briefly, plates (Nunc MaxiSorp, Thermofisher Scientific) were coated with 250 ng/ml SARS-CoV-2 S protein (SinoBiological) overnight or 300 ng/mL ORF8 (Masashi Mori, Ishikawa University, Japan) at 37 °C for 2 hours[20,70]. The plates were blocked with 1% FBS in PBS for 1 hour, then incubated with 1:100 heat-inactivated (HI) serum diluted in 0.05% Tween-20/ 0.1% FBS in PBS for 2 hours at room temperature before rinsing again. To assess antibody avidity, plates were washed 3 times with 8 M Urea before incubation for 2 hours with IgG-HRP (1:5000; G8-185, BD #555788). HRP was revealed by stabilised hydrogen peroxide and tetramethylbenzidine (R&D systems) for 20 min, stopped with 2 N H2SO4 and analysed with an absorbance microplate reader at 450 nm wavelength (Tecan Life Sciences). The IgG avidity index was given by the ratio of the OD450 values post-washing to pre-washing of the plates, which was only calculated when associated with a positive S IgG value and this was censored at 100%. To measure FcγRIIIa-binding antibodies, plates were instead coated with 500 ng/mL S protein, incubated with HI serum at 1:50 dilution for 1 hour at 37 °C and then with biotinylated FcγRIIIa-V158 developed in-house at 100 ng/ml for 1 hour at 37 °C. Streptavidin-HRP (1:10,000, Pierce) was then used to detect presence of S specific FcγRIIIa-V158-binding antibodies. OD450 values at or above the respective limits of detection (LODs) were considered positive, and values below were imputed as 0.5 of the LOD.

*T cell responses.* Peripheral blood mononuclear cells (PBMCs) were isolated from whole blood by density gradient separation then frozen in liquid nitrogen until use. For this T cell functional assay, thawed PBMCs were rested for 2 hours in 10% human AB serum supplemented RPMI medium. Next, the cells were stimulated with DMSO or 1 µg/mL overlapping peptide pools representing the SARS-CoV-2

S1/S2 subunits (StemCell Technologies, Vancouver, Canada), nucleocapsid (N) or membrane (M) proteins (Miltenyi Biotec, Bergisch Gladbach, Germany) for 16 hours in the presence of 1 μg/mL anti-CD28 and anti-CD49d costimulatory antibodies (clones CD28.2 and 9F10, Biolegend, San Diego, USA, #302902 and #304302). After 2 hours of stimulation, 10 μg/mL brefeldin A (Sigma, Kawasaki, Japan) was added[71]. The cells were then washed and subjected to immunostaining using a fixable viability dye (eBioscience, Santa Clara, USA, #65-0866-14, 1:60) and antibodies against CD3$^+$ (HIT3a, 1:60, #300318), CD4$^+$ (OKT4, 1:60, #317429), CD8$^+$ (HIT8a, 1:60, #300924), IFN-γ (B27, 1:15, #506507) and IL-2 (MQ1-17H12, 1:15, #500304) antibodies (Biolegend, San Diego, USA). Data acquisition was carried out using flow cytometry (LSR II with FACSDiva version 8.0; BD Biosciences, Franklin Lakes, USA) and analysed by Flowjo version 10 software (BD, Ashland, USA). Gating strategy is exemplified in Supp. Fig. 5. The antigen-specific T cells were calculated by subtracting the background (DMSO) data[72]. T cell response was considered positive when the frequency of cytokine-expressing cells was higher than 0.005% and the stimulation index was higher than 2. Negative values were imputed as 0.0025%.

**Outcomes**. The primary immunogenicity outcomes in this interim analysis included: S-specific antibody markers, which were the S IgG and S-RBD IgG levels, sVNT %inhibition, 90% and 50% PRNT titres, S IgG avidity and FcγRIIIa-binding; S-specific (and N- and M-specific for CC) IFN-γ$^+$ and IL-2$^+$ CD4$^+$ and CD8$^+$ T cell responses measured by the flow-cytometry-based intracellular cytokine staining assay; at 21 days post-dose 1 (or 28 days for CC) and 28 days after 2 doses at a prime-boost interval of 21 days (for BB) or 28 (for CC). The primary reactogenicity outcomes were solicited ARs and anti-pyretic use for 7 days after each vaccine dose.

Secondary immunogenicity outcomes included N and N-CTD IgG levels in CC recipients. For safety, the secondary outcomes were unsolicited AEs reported 28 days after each dose and SAEs collected throughout the study period. Other secondary outcomes not included in this interim analysis, such as the evaluation of similar outcomes in participants with severe paediatric illnesses, can be found in the Protocol and Statistical Analysis Plan (Supplementary Information).

### Statistical analyses

*Sample size and power estimation*. Power analyses were performed using G*Power (Heinrich-Heine-Universität Düsseldorf, Düsseldorf, Germany) and Sampsize (sampsize.sourceforge.net). For primary immunogenicity objectives, when comparing the peak geometric mean (GM) immunogenicity outcomes of children with that of adults, or between vaccine types, a sample size of 61 in each group would assure that a two-sided test with α = 0.05 has 99% power to detect an effect size with a Cohen's d value = 0.78, or a difference of 0.51 after natural log transformation, between 2 groups and a standard deviation (SD) of 0.65 on the natural log scale within each group[73,74]. For assays with higher technical requirements such as PRNT, 66 evaluable adolescents and 16 evaluable adults tested would achieve 80% power to detect the same difference with the same α and SD. For the proportion of participants with a positive result in immunogenicity outcomes or ARs, 110 adolescents would yield a 95% chance to detect the true value within ±7.5 percentage points of the measured percentage, assuming a prevalence of 80%. Recruitment of 120 adolescents were targeted per vaccine type to accommodate for attrition.

*Analysis populations*. The primary analysis of humoral and cellular immunogenicity outcomes was performed in healthy participants on a per-protocol basis. The evaluable analysis population included participants who were uninfected during the first 3 study visits (based on clinical history, baseline S-RBD IgG negativity, N IgG negativity for B or BB, ORF8 IgG negativity), generally healthy with no major protocol deviations, blood sampling within the evaluable window for post-dose 1 (no more than 3 days earlier or later than day 21 for B or day 28 for C, and before dose 2) or post-dose 2 timepoints (within day 14-42 post-dose 2 and before any further doses), and had a valid result for the relevant analysis and timepoint (see protocol in Supplementary Information). The expanded analysis population included similar criteria as the evaluable analysis population except the notable differences of the requirement of a valid immunogenicity result for the particular analysis at least 14 days post-dose 1 but before dose 2 and between 7-56 days post-dose 2 (see protocol in Supplementary Information). The non-inferiority hypothesis testing for primary immunogenicity outcomes included participants aged ≥18 years in the adult group and 11-17 years in the adolescent group.

*Statistical tests*. GMs were calculated for each immunogenicity outcome, timepoint and subgroup. GM ratios (GMRs) were calculated as exponentiated differences between the means of the natural log-transformed immunogenicity outcomes in the adolescent group and adult reference group. The GMRs were reported with a two-sided 95% CI for testing the non-inferiority hypothesis at the criterion of 0.60, with the basis of WHO criterion of 0.67 balanced by a pragmatic approach to achieve rapid delivery of study results[73,75,76]. Non-inferiority analyses were repeated in the expanded analysis populations, which also included more participants in a broader dosing and blood sampling intervals. Analysis subgroups were considered superior if the lower bound of the 95% CI for GMR with the comparator was >1 or inferior if the upper bound was <1[73,76]. The proportion of participants in each subgroup with a positive result (at or above the LOD, LOQ or cut-off) for a

test at a particular timepoint was reported in percentage with a 95% CI calculated by the Clopper-Pearson method. Comparisons of proportions were performed by the Fisher exact test. Immunogenicity outcome data below the cut-off were imputed with half the cut-off value. Comparisons of immunogenicity outcomes between groups were made by unpaired t test after natural log transformation.

As a secondary immunogenicity analysis, correlations between primary immunogenicity outcomes were evaluated by Pearson correlation coefficients after natural log transformation, with a more stringent significance level of $P = 0.01$ to account for multiple comparison testing. Relationships between sVNT %inhibition and baseline variables such as age, sex and haematological parameters were explored by multiple linear regression post-dose 1 by vaccine type in the adolescent group only at $P = 0.01$ arbitrarily for the multiple comparison adjustment.

Reactogenicity and safety analyses were conducted in healthy, uninfected participants who reported any safety or AR data in the adolescent group, and these comprised the healthy safety population. For the primary reactogenicity analysis, the proportion of types and severity of solicited ARs and antipyretic use within 7 days post-doses 1 and 2 are presented in percentages with the 95% CI calculated by the Clopper-Pearson method. The presence of each AR (regardless of severity) and antipyretic use was compared between vaccine types by the Fisher exact test. Incidences of AEs and SAEs reported by the 3rd study visit (28 days post-dose 2) are shown as a total number and events-per-participant by vaccine type.

As a secondary objective, vaccine efficacies (VEs) were estimated by correlation with neutralising antibody titres as previously established[12]. The mean neutralising levels (fold of convalescent) were derived by dividing the GMTs of PRNT90 in healthy evaluable adolescents receiving B, BB and CC with that of 102 convalescent sera collected on days 28-59 post-onset of illness in patients aged ≥18 years[38,39]. A single point estimate of VE was obtained for each vaccination by extrapolating the best fit of the logistic model, which was generated using an online plot digitizer tool (https://automeris.io/WebPlotDigitizer/, version 4.5).

Data analysis and graphing were performed on GraphPad Prism (version 9.3.1).

**Reporting summary**. Further information on research design is available in the Nature Research Reporting Summary linked to this article.

## Data availability

The study's Protocol and Statistical Analysis Plan are contained in the Supplementary Information. The data are available under restricted access for reasons of provision of interim analysis only and to protect the confidentiality of participants, and therefore deidentified participant-level datasets will be shared to researchers who provide a scientifically valid proposal. Since this study is ongoing, data will be available upon request 1 month after the completion of the study (anticipated in 2025). Access can be obtained by contacting lauylung@hku.hk.

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

## Acknowledgements
We thank the personnel from the HKUMed CVC at Ap Lei Chau Sports Centre and Gleneagles Hospital HK, particularly Dr Victoria WY Wong of HKU, Ms Cindy HS Man and Dr Hon-Kuan Tong who operated the territory-wide vaccination programme at the CVC and assisted in clinical monitoring of vaccine recipients alongside our group. The investigators are grateful to all clinical research team members and laboratory staff of Department of Paediatrics and Adolescent Medicine, including Mr KW Chan and Dr Davy CW Lee, for their research support. We are thankful to Dr Pamela P Lee at HKU and Dr Kai Ning Cheong at Hong Kong Children's Hospital who gave input to the study. We are indebted to the participants who have placed their trust in the investigators and joined the study. This study was supported by the research grant COVID19F02 awarded to Y.L. Lau by the Food and Health Bureau of the Government of Hong Kong, which was not involved in the study design, data collection, laboratory assays, statistical computation, interpretation or final conclusions of this project.

## Author contributions
Y.L.L. conceptualised the study. Y.L.L., M.P., W.T., W.H.L., D.L., J.S.R.D. and X.W. designed the study. Y.L.L. led the acquisition of funding. Y.L.L., W.T., M.P. and S.V. supervised the project. S.M.C., D.L., X.W., X.M., S.M.S.C., J.H.Y.L. led the study administrative procedures. W.H.S.W. provided software support. S.M.C., W.W.Y.T., P.I. and W.H.S.W. contributed to recruitment of participants. Y.L.L., J.S.R.D., and G.T.C. provided clinical assessments and follow-up. D.L., S.C., J.H.Y.L., J.S.R.D., Y.L.L. and G.T.C. collected safety data. S.M.S.C., S.C., K.K.H.K., K.C.K.C., J.K.C.L., L.L.H.L., L.C.H.T. and M.P. developed and performed S-RBD IgG, N IgG, N-CTD IgG, sVNT and neutralisation antibody assays. C.A.C., A.H., N.K. and S.V. developed and performed the S IgG, IgG avidity, S IgG Fcγ receptor IIIa-binding and ORF8 antibody assays. The specialised ORF8 protein was developed and provided by M.M., X.W., X.M., Y.Z. and W.T. developed and performed the T cell assays. D.L., J.H.Y.L., C.H.C., T.K.H. and W.H.S.W. curated the data. D.L., J.H.Y.L., J.S.R.D. and W.H.S.W. analysed the data. D.L., J.S.R.D., S.M.S.C., Y.L.L. and M.P. performed the vaccine efficacy extrapolation. D.L. and J.S.R.D. visualised the data. J.S.R.D., D.L., X.W., S.M.S.C., C.A.C., W.H.S.W., J.H.Y.L. and S.M.C. validated the data. J.S.R.D. and D.L. wrote the first draft as supervised by Y.L.L., with input from X.W., S.M.S.C. and C.A.C. All authors reviewed and approved the final manuscript.

## Competing interests
The authors declare no competing interests.

## Additional information

**Peer review information** *Nature Communications* thanks Naor Bar-Zeev and the other anonymous reviewers for their Contribution to the Peer review of this work. Peer review reports are available.

