## [Peer review file · Nature Communications]

REVIEWER COMMENTS

Reviewer #1 (Remarks to the Author):

Overall and Title:

The authors review and compare humoral and cellular immune responses of adults and adolescents to the BNT162b2 (1 or 2 dose) and CoronaVac (Sinopharm, 2 dose) vaccine regimens. I recommend that the authors consider specifying these two vaccines in the title, rather than using the more general “mRNA and inactivated vaccines” language, since the manuscript limits its review to these two singular examples of each vaccine type.

I have offered some recommendations for the authors' consideration below, but overall, the work is comprehensive, well done and well-written, and merits timely publication.

Introduction:

Line 104: Recommend that the authors add “in adults” to clarify the population for which the WHO review and emergency use determination applied, particularly in this paper focused on a comparison of adolescents and adults.

Line 112: The authors should consider adding a reference to Gilbert P, Montefiori D, McDermott A, et al, (Science, 375, p43-50 (23 Nov 2021) in light of its contribution to mRNA CoP specified by the authors here (i.e., nAb and S IgG)

Lines 115-117: This sentence follows a sentence (in Lines 112-115) that asserts that protection may be due to immune responses other than binding or neutralizing antibodies, but it then repeats the potential role of non-neutralizing antibodies—thus, this sentence (as currently worded) does not provide an example of something other than nAb or bAb contribution to protection. Reference 17, however, expounds upon the potential role of T-cell responses, as well, so perhaps this sentence should be removed, or re-worded and moved to the following paragraph on the potential contribution of T-cell responses to protection.

Lines 132-134: It is not clear that this sentence belongs here in a paragraph about the contributions of the T-cell response, rather than a humoral response, to protection. A rapid response at the nasal mucosa to prevent acquisition of asymptomatic infection may not rely primarily or exclusively on T-cell responses but, instead, on the presence of nAbs, as reported, for example, in pre-exposure prophylaxis monoclonal antibody trials (e.g., O'Brien M, et al, NEJM 2021; 385:1184-1195 & O'Brien MP et al, JAMA 2022; 327(5):432-441). Recommend that the authors clarify further, or delete.

Results:

Extended Data Fig 1 and Lines 168-178: With the current format of the Extended Data Figure, and text descriptions of it, it is difficult to understand which participants were included in which analyses reported here. For example, the figure but not the text references visit numbers, and the various cohorts appear to go by multiple different names (e.g., mITT or expanded analysis population, etc). Please clarify, using consistent language throughout figure(s) and text and clearly, consistently naming the cohorts or sub-cohort analysis populations throughout.

Extended Data Fig 1: A large proportion (over 35%) of adults receiving the CC vaccination regimen had blood sampling outside the evaluable interval (eg, 2-6 weeks post-dose 2)—many times more than the loss of evaluable samples from adolescents after CC vaccination. It would be appropriate for the authors to acknowledge this considerable loss of sample size and provide the reasons for it, among other reasons to help reassure the reader(s) that no potential bias is introduced into the results from such a significant and disproportionate loss of sample from one group that is then compared to another.

Lines 196-199: The difference in N and N-CTD IgG seropositivity rates across adolescents vs. adults is one of the more prominent and interesting distinctions observed in the trial and merits more exploration in the Discussion, including consideration of potential mechanisms and implications. Those with prior infection were excluded, but could the relatively wide range of allowable sample collection time points (per Line 588 in Methods, 14-42 days post-dose 2 in primary analysis) have contributed to this difference if, for example, mean sampling time point for adults and adolescents were considerably different?

Lines 215-218: Please clarify whether these levels of N and N-CTD IgG were compared across responders only, or across all (noting the considerably lower response rate among adults compared to adolescents as described in Lines 196-199)

Line 239: “T cell responses to S after B and to...” I believe should also add “T cell responses to S after B or BB or CC and to...”

Line 311-12: It seems curious that there was no correlation between humoral and cellular outcomes, e.g., between CD4+ T cell responses and antibody responses. The manuscript may benefit from the authors’ speculation on the reason(s) for this. Perhaps related to response kinetics not observed in the sampling time frame reported here? This could be added to the Discussion.

Lines 312-315: Were the associations explored here only explored after one dose (i.e., “after B or C” as written in Line 315) or after the 2-dose regimen? Did the authors also explore associations with co-morbidities, e.g., diabetes, elevated BMI or other factors well described in the literature as impacting immunologic profiles in response to infection?

Figure 1: This is a very nice figure, with much substantive information very well-conveyed. I don’t see a note re: how the non-inferiority margin (0.6-1) was chosen; please add (eg, to Methods), unless I missed it.

Discussion:

Line 337: “titres was non-inferior” should be “titres were non-inferior”

Line 348: Consider specifying CD8+ T cells (as a defence against disease progression).

Line 357-8: specify “...antibody titres, SARS-CoV-2 vaccines of different platforms...”

Lines 347-362: The authors should consider expanding this slightly with additional references (e.g., Keeton R et al, Nature, 603, p488-492 [2022], and others that the authors have also used later in the manuscript) exploring the potential contribution of T-cell responses for COVID-19, particularly considering that in Lines 360-361, they provide a single (insufficient) reference for “multiple lines of evidence” supporting the role of T cell responses in mediating severe COVID-19 outcomes.

Lines 395-7: This statement would be stronger if it were expanded to include references beyond this singular—but good—example from South Africa (e.g., examples from England, Denmark, etc are now available). There are particularities re: the South African experience—ie, a young demographic, high baseline seroprevalence, etc—that may not necessarily translate more globally, so this statement would be strengthened by supplementing the RSA experience referenced here with additional examples, as well.

Line 403-4: This is a stronger final sentence. The remainder of this paragraph raises a new topic— intradermal route of administration—that is interesting to consider but would make more sense if raised earlier in the Discussion. If the authors choose to raise this topic earlier in the Discussion, it could be helpful for them to address whether the inactivated influenza example raised here may be particularly applicable to humoral responses (ie, in Line 406 enhancing humoral responses, specifically) and how the referenced trial in neonates and infants (another unique immune milieu may (or may not) translate to adolescents and adults, which are the focus of the remainder of this paper.

Methods:

The non-blinded and non-randomized design is a serious consideration that could particularly undermine the veracity of their safety/reacto assessments and comparisons, particularly given the considerations already acknowledged by the authors with respect to the well-publicized safety profile of BNT162b2 among adolescents (e.g., pain, myocarditis). However, I think the authors adequately acknowledge this limitation of their study design and their reporting of the safety profiles and comparisons is fair.

Line 424: How was “known history of COVID-19” identified? This would be helpful to specify here, whether by potential participant self-report, baseline serologic testing, or otherwise.

Lines 588 and 593: The windows for post-dose 2 sampling timepoints are somewhat large—2-6 weeks in the “evaluable analysis population” or 1-8 weeks in the “expanded analysis population.” The need for wide windows in such trials is understandable, but can also significantly impact immunologic parameter measurements, particularly in such early post-vaccination weeks. Thus, the authors should clearly characterize this variability across the subgroups (eg, reporting mean post-dose 2 sampling time points and SD adolescents and adults in each vaccine regimen subgroup) and control for it in the comparative analysis of adolescent and adult titres, response rates, etc.

Reviewer #2 (Remarks to the Author):

This is a fantastic paper, beautifully and clearly written, and thoughtfully conducted.

Have been waiting for data such as these, so it is wonderful to see. The policy relevance of the findings is well articulated in the discussion, which also appropriately addresses the question of omicron specific responses, a question which is begged throughout a reading of this paper. I do have (quite) a few questions though, but none of these detracts from the high quality and importance of this paper.

line 129 I believe studies of T cell responses have also reported on preserved longevity of response. This is not mentioned in the introduction, but the authors do mention in in the discussion in passing with respect to HBV. I also wonder whether time since vaccine could be a factor for future analysis?

line 221 The non-inferiority margin is 0.6. It is not stipulated if this is on a relative scale. The clinical or immunological justification for this margin is not given anywhere in the manuscript. And 0.6 seems rather a large margin. However in line 122 "GMR 0.48, 95% 0.42-0.54" is reported to fail non-inferiority, that is, there is evidence the result is inferior. But this is confusing since 0.48 is within the 0.6 margin. Could this be made clearer throughout?

line 224 "were all significantly lower" seems no longer to be reporting on non-inferiority, but rather on the usual superiority (here inferiority) approach. Is this right?

line 286/7 CoP and PRNT90. I am uncertain if the Khoury paper compared any assay in vaccinated vs the same assay in convalescent sera, or whether their curve was normed on PRNT50. Their text implies that might be the case. In any event, our authors performed both PRNT50 and PRNT90 (as stated in line 221), so could they address their use of PRNT90 for the CoP evaluation? (As an aside, I commend the authors for setting their own bar higher with the decision to evaluate PRNT90.)

line 330 The argument in this sentence is unclear. The logical conclusion in the sentence (which is presented in the first half) does not follow from the premise in the sentence (presented in its second half). Why would preventing hospitalization in the young bring the pandemic to endemicity?

line 415-416 The implied primary objective here is to compare across two vaccines in children. But in lines 157-158 the authors state the actual aim of noninferiority of responses in adolescents compares to adults. Seems that lines 415-6 should be revised.

line 422 Were the recruited adults related to the participating children? Line 569 says "comparing... of children with that of parents". If so, this would represent a convenience sample, but raise some issues regarding lack of independence between the groups, particularly with respect to presumed biological similarity which could bias towards a traditional null (though here toward non-inferiority), or at least imply a need for paired analyses (like McNemar tests where the authors used Fisher's exact test?) Actually I am quite unsure how one would deal with a non-inferiority framed question in the context of clustered non-independent responses. Please would the authors clarify?

line 424 "known history of COVID-19" the authors don't explain how this was ascertained, was it by self report? or documentation? Was baseline serology done? Though in line 585 the authors report collecting clinical history, baseline S-RBD IgG negativity, N and ORF8 IgG negativity. And on line 271 the authors report on baseline T cell responses. So which of all these was used to establish this exclusion criterion?

lines 468, 505, 506, 525, 548x, 610: imputation of half the detectable lower limit was used widely in this work. How prevalent was non-detection? This should be reported, since it can introduce bias, especially for outcomes that are reported as fold- change (line 631).

line 485 and 499. Was the pseudovirus that was used based on Wuhan parental strain? The authors mention in the discussion that omicron was not used. But please clarify the strain that was in fact used, in terms of its relation to dominant global variants.

line 548 were any T cell functional assays performed, like proliferative responses or the like? Is this relevant?

line 569 - see line 422 above

line 570 two-tailed tests are mentioned. But this is unusual for a non-inferiority design.

line 577 When the authors state 7.5% do they mean 7.5 percentage points? Is a relative or absolute scale being used?

line 603-604 is a bit confusing, since it seems to imply a bioequivalence approach and not a non-inferiority approach.

line 614 good to see a smaller alpha given multiple comparisons. But why was $P=0.01$ chosen? Arbitrarily or based on some specific consideration?

line 634-635 Estimating VE based on immune response curves, though an important advance during the pandemic, is still no replacement for epidemiologically rigorous VE studies. So at very least the authors should address the caveats of this.

Linguistic and other pedantry:

line 85 "in more adolescents than adults", unless denominators are equal, this statement is unclear. Perhaps the authors mean "more so in adolescents than in adults"

line 104 "and effective" should be "and efficacious", since WHO EUL was on basis of RCT results.

line 163 "of which 646 who provided consent"  "of whom 646 provided consent"

line 214 "and thus non-inferior and superior for adolescent CC" - this sentence is unclear. Was it cut off in error?

line 226 "Despite satisfying non-inferiority for S-RBD IgG and sVNT, adolescent B was indeed inferior to adult BB on both tests" Suggest rephrase "Despite BB satisfying non-inferiority for S-RBD IgG and sVNT, adolescent B was indeed inferior to adult BB on both tests." (As a minor aside - failing non-inferiority does not make something "indeed inferior", it just makes it not non-inferior. Under Popperian principles, the null hypothesis can never be proven, it can only be rejected. But biologically stating this is inferior is indeed sensible, statistical technicalities notwithstanding.)

line 525 The segment commences on avidity, then discussion Fc-R binding, then concludes with avidity again. Is this intentional? Should this be restructured, or does the final section refer also to the Fc-R binding assay?

Reviewer #3 (Remarks to the Author):

The authors conducted a carefully designed head-to-head comparison of BNT162b2 and CoronaVac in adolescents with a remarkable and comprehensive characterization of immune response. In addition, adults receiving both vaccines were also enrolled to immunobridge between the two populations.

Higher anti-spike serological readouts were observed in adolescents receiving 2 doses of BNT162b2 than CoronaVac, while similar T cell responses were observed. The readouts in adolescents were not inferior to those in adults, supporting the use of CoronaVac in adolescent population in absence of efficacy data in regions where mRNA vaccines are not available. Furthermore, the authors analyzed immune readouts in adolescents receiving 1 dose of BNT162b2, which was inferior to those in adults receiving 2 doses of the same vaccine.

Two minor points:

1. A supplemental table listing the dosage of both vaccines in adolescents and adults will be very helpful.
2. A more detailed writeup of the PRNT assay, or a reference (if there is one) will be helpful.

REPLY TO REVIEWERS

We, the authors, greatly appreciate the constructive comments from the reviewers that have helped improve this manuscript. All comments/suggestions by the reviewers have been addressed in the revised version of the main manuscript, which are referred in our responses below for convenience to the reviewers.

REVIEWER COMMENTS TO AUTHORS:

Reviewer: 1

The authors review and compare humoral and cellular immune responses of adults and adolescents to the BNT162b2 (1 or 2 dose) and CoronaVac (Sinopharm, 2 dose) vaccine regimens. I recommend that the authors consider specifying these two vaccines in the title, rather than using the more general “mRNA and inactivated vaccines” language, since the manuscript limits its review to these two singular examples of each vaccine type.

I have offered some recommendations for the authors' consideration below, but overall, the work is comprehensive, well done and well-written, and merits timely publication.

We thank the reviewer for this kind summary and appreciation of our work. We have changed the manuscript's title to be more specific about the vaccines as suggested by the reviewer.

(Title, Page 1, lines 1-2)

Immunogenicity and reactogenicity of SARS-CoV-2 vaccines BNT162b2 and CoronaVac in healthy adolescents

Line 104: Recommend that the authors add “in adults” to clarify the population for which the WHO review and emergency use determination applied, particularly in this paper focused on a comparison of adolescents and adults.

The reviewer is correct that BNT162b2 was initially recommended for ≥ 16 years old and CoronaVac for ≥ 18 years old only during the initial phase. We thank the reviewer for the recommendation to specify more clearly the population as discussed. This has been added to the revised version of the manuscript accordingly.

(Introduction, Page 4, line 104)

Several vaccines against SARS-CoV-2, utilizing novel nucleoside-modified mRNA technologies and the conventional inactivated whole-virus platform, underwent an expeditious review process by the World Health Organization (WHO) and were deemed adequately safe and efficacious for emergency use in adults during the initial phase.^{5,6}

Introduction:

Line 112: The authors should consider adding a reference to Gilbert P, Montefiori D, McDermott A, et al, (Science, 375, p43-50 (23 Nov 2021) in light of its contribution to mRNA CoP specified by the authors here (i.e., nAb and S IgG).

We thank the reviewer for this recommendation and have added this important reference accordingly.

(Introduction, Page 4, line 113)

*Vaccine efficacy has been linked to markers of immunological response known as correlates of protection (COP) in many infectious diseases, most commonly neutralizing antibody titres.*¹¹⁻¹⁴

(References, Page 30, reference 14)

14. Gilbert, P.B., et al. Immune correlates analysis of the mRNA-1273 COVID-19 vaccine efficacy clinical trial. *Science* 375, 43-50 (2022).

Lines 115-117: This sentence follows a sentence (in Lines 112-115) that asserts that protection may be due to immune responses other than binding or neutralizing antibodies, but it then repeats the potential role of non-neutralizing antibodies— thus, this sentence (as currently worded) does not provide an example of something other than nAb or bAb contribution to protection. Reference 17, however, expounds upon the potential role of T-cell responses, as well, so perhaps this sentence should be removed, or re-worded and moved to the following paragraph on the potential contribution of T-cell responses to protection.

We thank the reviewer for this suggestion on rewording to better convey this message. We have modified accordingly in the revised version of the manuscript. Of note, this previous reference 17 (now reference 18) expounded on the importance of non-neutralizing antibodies (this paragraph) and T cell responses (next paragraph), and therefore we used this reference for this paragraph that discusses non-neutralizing antibodies and the next paragraph that discusses T cell responses.

(Introduction, Page 4, lines 111-116)

*Vaccine efficacy has been linked to markers of immunological response known as correlates of protection (COP) in many infectious diseases, most commonly neutralizing antibody titres.*¹¹⁻¹⁴ *However, host defence against the viral infection involves many constituents of the immune system acting synergistically and dynamically, rather than merely reflected by antibody neutralization.*^{15,16} *As examples, spike protein (S) IgG are also a COP against symptomatic COVID-19, and the onset of efficacy in mRNA vaccines coincides with the presence of binding antibodies and precedes neutralizing antibody production.*^{17,18}

(Introduction, Page 5, line 125)

To prevent progression to severe illness, T cells also play a major role in orchestrating a focused spectrum of immune responses, such as directing apoptosis of infected cells and antibody germinal centre reactions for high-avidity class-switched responses.^{18,23-25}

Lines 132-134: It is not clear that this sentence belongs here in a paragraph about the contributions of the T-cell response, rather than a humoral response, to protection. A rapid response at the nasal mucosa to prevent acquisition of asymptomatic infection may not rely primarily or exclusively on T-cell responses but, instead, on the presence of nAbs, as reported, for example, in pre-exposure prophylaxis monoclonal antibody trials (e.g., O'Brien M, et al, *NEJM* 2021; 385:1184-1195 & O'Brien MP et al, *JAMA* 2022; 327(5):432-441). Recommend that the authors clarify further, or delete.

We thank the reviewer for this input. We have deleted this sentence as suggested in the revised manuscript.

(Introduction, Page 5)

*Protection against asymptomatic infection likely involves immunity at the respiratory mucosa rather than peripheral blood, as a potent response is likely required to clear offending viral seeding rapidly.*³⁴

Results:

Extended Data Fig 1 and Lines 168-178: *With the current format of the Extended Data Figure, and text descriptions of it, it is difficult to understand which participants were included in which analyses reported here. For example, the figure but not the text references visit numbers, and the various cohorts appear to go by multiple different names (e.g., mITT or expanded analysis population, etc). Please clarify, using consistent language throughout figure(s) and text and clearly, consistently naming the cohorts or sub-cohort analysis populations throughout.*

We thank the reviewer for this suggestion, and we have more clearly described with consistent language, such as the use of expanded analysis population only, in Extended Data Fig 1 and the text of the revised manuscript.

(Results, Page 7, lines 164-181)

Enrolment of study participants. *A total 658 volunteers were screened, of whom 646 provided consent, consisting of 309 adolescents and 337 adults at dose 1, respectively, were enrolled between 27 April 2021 and 23 October 2021 (see Methods; Extended Data Fig. 1). Based on clinical history and serological screening, 26 were enrolled in separate prior COVID-19 and 93 in immune/paediatric diseases sub-studies. This present interim analysis of our study that will track severe adverse events and immunogenicity outcomes over a 3-year period focused on healthy participants, consisting of 239 adolescents (11-17 years old, mean=14.0, SD=1.7) and 288 adult (18-67 years old, mean=47.5, SD=7.5) participants (total N=527) who received at least 1 dose of either BNT162b2 or CoronaVac. They all returned for a subsequent follow-up visit (Visit 2) and were included in the reactogenicity and safety (healthy safety population; see Methods, and Protocol and Statistical Analysis Plan in Supplementary Information) analyses. Similar numbers completed the 2-dose (BB for BNT162b2 and CC for CoronaVac) vaccination series and returned for the subsequent follow-up visit (Visit 3) (Extended Data Table 1). Demographic characteristics (Extended Data Table 1) and the mean time of blood sampling (data not shown) were evenly distributed. The evaluable analysis population in this analysis included those uninfected as assessed at any study visits, with no major protocol deviations and had a valid immunogenicity result (see Methods; Extended Data Fig. 1). There were 223 adolescents and 166 adults in the evaluable analysis population for primary immunogenicity after 2 doses. We confirmed the findings using a secondary analysis in an expanded analysis population, consisting of 226 adolescents and 223 adults that had relaxed vaccination and blood taking intervals (see Methods; Extended Data Fig. 1).*

(Extended Data Fig. 1, line 8)

The evaluable analysis population included 223 adolescents and 166 adults for the primary immunogenicity outcomes after 2 doses. For the corresponding expanded analysis, there were 226 adolescents and 223 adults.

Extended Data Fig 1: A large proportion (over 35%) of adults receiving the CC vaccination regimen had blood sampling outside the evaluable interval (eg, 2-6 weeks post-dose 2)—many times more than the loss of evaluable samples from adolescents after CC vaccination. It would be appropriate for the authors to acknowledge this considerable loss of sample size and provide the reasons for it, among other reasons to help reassure the reader(s) that no potential bias is introduced into the results from such a significant and disproportionate loss of sample from one group that is then compared to another.

We thank the reviewer for the concern of potential bias due to adult CCs who were out of the evaluable analysis population. We have provided the explanation for this in the revised manuscript as follows:

(Discussion, Page 17, lines 404-407)

Thirty-three adults received CC but did not have blood sampling within the evaluable interval, since CoronaVac has been available to adults in Hong Kong for 13-14 weeks prior to study initiation. However, non-inferiority testing in the evaluable and expanded analysis populations yielded consistent findings, ensuring the validity of our results.

Lines 196-199: The difference in N and N-CTD IgG seropositivity rates across adolescents vs. adults is one of the more prominent and interesting distinctions observed in the trial and merits more exploration in the Discussion, including consideration of potential mechanisms and implications. Those with prior infection were excluded, but could the relatively wide range of allowable sample collection time points (per Line 588 in Methods, 14-42 days post-dose 2 in primary analysis) have contributed to this difference if, for example, mean sampling time point for adults and adolescents were considerably different?

We agree and thank the reviewer for this astute appreciation that the N and N-CTD IgG seropositivity findings are prominent and interesting. Data on the mechanisms and implications of N and N-CTD IgGs for COVID-19 are scarce and contained in the reference that was already included. Otherwise, only very basic knowledge regarding the molecular function is known, which we added in the Discussion.

The reviewer brought up a reasonable possibility that the range of sample collection time points as a contributor to this observation. The mean, standard deviations (SD) and differences (Diff) of time of blood sampling in days between visit 3 (post-dose 2) and visit 2 (dose 2) for adolescents and adults and for each vaccine are shown in the table below (the limit of tables and figures for this journal have been reached). The blood sampling time differences of <3 days in the evaluable groups and <10 days in the expanded groups were not expected to result in clinically meaningful differences in N and N-CTD IgG levels.

These points have been added in the revised manuscript.

	BNT162b2 Adolescents		BNT162b2 Adults		Diff	CoronaVac Adolescents		CoronaVac Adults		Diff
	Mean	SD	Mean	SD		Mean	SD	Mean	SD	
Evaluable	28.5	5.1	31.0	2.4	2.5	28.2	5.2	30.4	2.2	2.2
Expanded	29.0	7.0	33.2	4.2	4.2	28.4	10.4	38.0	2.8	9.6

Numbers are in days

(Results, Page 7, lines 174-175)

Demographic characteristics (Extended Data Table 1) and mean time of blood sampling (data not shown) were evenly distributed.

(Discussion, Page 16, lines 384-389)

Additionally, inhibitors of GSK-3 that activates N impair SARS-CoV-2 replication in lung epithelial cells.⁵⁹ The clinical relevance of N is illustrated by the observation that patients taking lithium, which has inhibitory activity against GSK-3, have reduced risk of COVID-19.⁵⁹ However, the underlying mechanisms and implications of the higher N and N-CTD IgGs differentially in adolescents in this study are unknown and deserve further research.

(References, Page 33, reference 59)

59. Liu, X., et al. Targeting the coronavirus nucleocapsid protein through GSK-3 inhibition. Proc Natl Acad Sci U S A 118(2021).

Lines 215-218: Please clarify whether these levels of N and N-CTD IgG were compared across responders only, or across all (noting the considerably lower response rate among adults compared to adolescents as described in Lines 196-199).

We thank the reviewer for seeking clarification on this important point, which we have provided in the revised manuscript.

(Results, Page 8, line 201)

N IgG and N-CTD IgG seropositivity in adolescent CC was high, at 98.4% and 92.2%, with GM OD450 of 1.72 and 2.09 across all responders and non-responders, while only 52.4% and 28.6% of 21 adult CC (GM OD450 of 0.77 and 0.92) selected at random were seropositive, respectively.

Line 239: “T cell responses to S after B and to...” I believe should also add “T cell responses to S after B or BB or CC and to...”.

We appreciate the reviewer for suggesting to add more clarity to the comparisons and have added this accordingly.

(Results, Page 10, lines 235-236)

Cellular immunogenicity outcomes between adolescents and adults. Interferon- γ (IFN- γ)⁺ and interleukin-2 (IL-2)⁺ CD4⁺ and CD8⁺ T cells responses specific to S after B or BB or CC (and to N and M for CC) were analyzed with intracellular cytokine staining on flow cytometry for 21-28 days post-dose 1 and 28 days post-dose 2 as primary outcomes (58 B, 56 BB and 60 CC evaluable adolescents were included; see Methods).

Line 311-12: It seems curious that there was no correlation between humoral and cellular outcomes, e.g., between CD4⁺ T cell responses and antibody responses. The manuscript may benefit from the authors’ speculation on the reason(s) for this. Perhaps related to response kinetics not observed in the sampling time frame reported here? This could be added to the Discussion.

We agree with this astute observation by the reviewer, and we are intrigued by the current findings of no correlation between CD4⁺ T cell responses and antibody responses. Several recent studies ([https://doi.org/10.1016/S0140-6736\(21\)01694-9](https://doi.org/10.1016/S0140-6736(21)01694-9);

<https://doi.org/10.1126/sciimmunol.abf3698>; [https://doi.org/10.1016/S2666-5247\(21\)00275-5](https://doi.org/10.1016/S2666-5247(21)00275-5)) have also found no or low correlations between antibody and T cell responses in infection and vaccination. This discordance highlights the dynamic complexity of our immune system and its vaccine responses. Indeed, it is possible that one reason for the current finding of no to low correlations could be related to the response kinetics not observed in the sample timeframe as studied. More research on T cell, B cell and antibody responses to COVID-19 vaccines are necessary to better understand the interplay of these pathways that result in the development of immunity to COVID-19 in humans, including adolescents. We have added this speculation in the revised manuscript as suggested by the reviewer.

(Discussion, Page 14, lines 332-336)

There was no or low correlations between antibody and T cell responses at the studied timepoint, and the reasons for this are unclear. One possibility is that the link between the two arms of the adaptive immune system is dynamic, and the observed discordance was possibly related to response kinetics not being observed at the sampling timeframe.

Lines 312-315: Were the associations explored here only explored after one dose (i.e., “after B or C” as written in Line 315) or after the 2-dose regimen? Did the authors also explore associations with co-morbidities, e.g., diabetes, elevated BMI or other factors well described in the literature as impacting immunologic profiles in response to infection?

We thank the reviewer for seeking clarification on these important points.

For Extended Data Fig. 4a and Extended Data Fig.4 b, most values of the studied immunogenicity outcomes especially sVNT were high after 2 doses of vaccines, with very few values that were zero or in between. As such, correlation results after 2 doses were not informative.

For Extended Data Table 5, this study focused on healthy participants as homogeneous groups, and therefore the adolescents in did not have co-morbidities, such as diabetes or other known immunocompromised conditions. We added weight for height (parameter for obesity in adolescents) into the multiple regression analysis (Extended Data Table 5) as suggested by the reviewer, which did not show a significant association. Of note, the independent variables, height and weight (previously with no signification association with the immunogenicity outcome), were removed from the multiple regression model since these would be overlapping variables to the newly added weight for height.

Clarification of these points have been added to the revised manuscript.

(Results, Page 13, lines 314-315)

Immunogenicity outcomes after 2 doses of vaccines were high for many participants, and therefore the correlation analyses were performed for post-dose 1 only.

(Results, Page 13, lines 320 and 322-323)

We also explored associations between sVNT levels and baseline demographic, anthropometric (including weight for height for age and sex) and haematological variables (total white blood cell count, absolute lymphocyte count, haemoglobin concentration) after B or C, which yielded no significant findings (Extended Data Tables 5 and 6). The participants were generally healthy without major associated co-morbidities.

Extended Data Table 5. Multiple regression analysis of baseline variables and sVNT percent inhibition.						
Variable	Adolescent B sVNT percent inhibition			Adolescent C sVNT percent inhibition		
	Estimate	95% CI (asymptotic)	P-value	Estimate	95% CI (asymptotic)	P-value
Age	0.0	-1.5, 1.6	0.98	-1.9	-3.9, 0.1	0.07
Male sex	0.5	-4.0, 5.0	0.82	-3.6	-12.2, 5.1	0.42
Han Chinese	-2.3	-8.2, 3.7	0.45	/	/	/
WBC	0.5	-0.99, 2.1	0.70	0.6	-3.3, 4.6	0.76
ALC	3.7	-1.2, 8.5	0.13	-3.4	-12.1, 5.3	0.44
Hemoglobin	-0.2	-2.5, 2.1	0.87	2.5	-0.8, 5.8	0.14
Weight for height	0.0	-0.1, 0.1	0.99	0.1	-0.1, 0.2	0.32

B, BNT162b2 1 dose; C, CoronaVac 1 dose; WBC, white blood cell count; ALC, absolute lymphocyte count.
Weight for height was the percentile for age and sex.

Figure 1: This is a very nice figure, with much substantive information very well-conveyed. I don't see a note re: how the non-inferiority margin (0.6-1) was chosen; please add (eg, to Methods), unless I missed it.

We thank the reviewer for requesting more information on how 0.6-1 was chosen, which was predefined in the manuscript's attached study Protocol and Statistical Analysis Plan that were completed prior to the data analysis step. This was described in the Analysis populations subsection of the Online Methods.

The World Health Organization has recommended 0.67 for non-inferiority studies of vaccines (reference below). This stringent margin is not definite, and another landmark COVID-19 vaccine study had chosen 0.63 “on a pragmatic basis to approach the WHO criterion of 0.67 for licensing new vaccines when using GMR as the primary endpoint, while still allowing rapid study delivery,” as example (reference below), which was also our aim, and we had rounded off to the nearest 2 decimal points of 0.60.

We added more clarification of these points in the revised manuscript. Additionally, to support the selection of these cut-offs that were used for the non-inferiority margin, we have added relevant references to the Analysis populations subsection of the Online Methods that describes this methodology.

(Online Methods, Pages 26-27, lines 628-634)

The GMRs were reported with a two-sided 95% CI for testing the non-inferiority hypothesis at the criterion of 0.60, with the basis of WHO criterion of 0.67 balanced by a pragmatic approach to achieve rapid delivery of study results.^{75,77,78} Non-inferiority analyses were repeated in the expanded analysis populations, which also included more participants in a broader dosing and blood sampling intervals. Analysis subgroups were considered superior if the lower bound of the 95% CI for GMR with the comparator was >1 or inferior if the upper bound was <1.^{75,78}

(References, Page 34, references 75 and 77-78)

75. Liu, X., et al. Safety and immunogenicity of heterologous versus homologous prime-boost schedules with an adenoviral vectored and mRNA COVID-19 vaccine (Com-COV): a single-blind, randomised, non-inferiority trial. *Lancet* 398, 856-869 (2021).

77. World Health Organization. Guidelines on clinical evaluation of vaccines: regulatory expectations. <https://www.who.int/publications/m/item/WHO-TRS-1004-web-annex-9> (2017).

78. Bikdeli, B., et al. Noninferiority Designed Cardiovascular Trials in Highest-Impact Journals. *Circulation* 140, 379-389 (2019).

Discussion:

Line 337: “titres was non-inferior” should be “titres were non-inferior”

We appreciate the reviewer detailed grammatical check and have modified in the revised manuscript accordingly.

(Discussion, Page 14, line 347)

The phase 2 licensing trial on CoronaVac did not formally test for non-inferiority; however, we also showed that PRNT GM titres were non-inferior, supporting the use of CoronaVac in adolescents in the absence of efficacy data.

Line 348: Consider specifying CD8+ T cells (as a defence against disease progression).

We have added CD8⁺ in this statement of the revised manuscript to be more specific as the reviewer advised.

(Discussion, Page 15, line 359)

We demonstrated similar or non-inferior T cell response in adolescent vaccinees. While a COP has yet to be confirmed for protection against severe COVID-19, CD8⁺ T cells are likely a key defence against disease progression.^{16,43}

Line 357-8: specify “...antibody titres, SARS-CoV-2 vaccines of different platforms...”

We appreciate the reviewer for suggesting to specify the “SARS-CoV-2” in this sentence and have added accordingly.

(Discussion, Page 15, lines 368-369)

Moreover, despite exponential differences in neutralizing antibody titres, SARS-CoV-2 vaccines of different platforms including mRNA, adenoviral vector and inactivated vaccines have been shown to produce potent T cell responses and very high effectiveness against hospitalization.^{31,47-49}

Lines 347-362: The authors should consider expanding this slightly with additional references (e.g., Keeton R et al, *Nature*, 603, p488-492 [2022], and others that the authors have also used later in the manuscript) exploring the potential contribution of T-cell responses for COVID-19, particularly considering that in Lines 360-361, they provide a single (insufficient) reference for “multiple lines of evidence” supporting the role of T cell responses in mediating severe COVID-19 outcomes.

We thank the reviewer for this suggested reference, which we have included in the revised manuscript in addition to the others we had also used later in the manuscript, as suggested by the reviewer.

(Discussion, Page 15, lines 371-372)

Multiple lines of evidence support the contention that T cells play a major role in mediating protection against severe COVID-19.^{31,49-57}

(References, Pages 32-33, references 51 and also inserted references 31, 49, 52-57)

51. Keeton, R., et al. T cell responses to SARS-CoV-2 spike cross-recognize Omicron. *Nature* **603**, 488-492 (2022).

Lines 395-7: This statement would be stronger if it were expanded to include references beyond this singular—but good— example from South Africa (e.g., examples from England, Denmark, etc are now available). There are particularities re: the South African experience— ie, a young demographic, high baseline seroprevalence, etc—that may not necessarily translate more globally, so this statement would be strengthened by supplementing the RSA experience referenced here with additional examples, as well.

We have the references from the UK, Denmark and Hong Kong in the revised manuscript as suggested by the reviewer.

(Discussion, Page 18, lines 423-424)

*This lasting cellular immunity is one of many possibilities that contributes to the high clinical effectiveness of 2 doses of BNT162b2 doses against hospitalization and death during the initial Omicron wave.*⁶⁶⁻⁷⁰

(References, Pages 33-34, references 66-68)

66. Collie, S., Champion, J., Moultrie, H., Bekker, L.G. & Gray, G. Effectiveness of BNT162b2 vaccine against Omicron variant in South Africa. *N Engl J Med* (2021).

67. UK Health Security Agency. COVID-19 vaccine surveillance report: Week 12. (2022).

68. Bager, P., et al. Reduced Risk of Hospitalisation Associated With Infection With SARS-CoV-2 Omicron Relative to Delta: A Danish Cohort Study. *SSRN* (2022).

69. Gram, M.A., et al. Vaccine effectiveness against SARS-CoV-2 infection and COVID-19-related hospitalization with the Alpha, Delta and Omicron SARS-CoV-2 variants: a nationwide Danish cohort study. *medRxiv*, 2022.2004.2020.22274061 (2022).

70. McMenamin, M.E., et al. Vaccine effectiveness of two and three doses of BNT162b2 and CoronaVac against COVID-19 in Hong Kong. *medRxiv* (2022).

Line 403-4: This is a stronger final sentence. The remainder of this paragraph raises a new topic—intradermal route of administration—that is interesting to consider but would make more sense if raised earlier in the Discussion. If the authors choose to raise this topic earlier in the Discussion, it could be helpful for them to address whether the inactivated influenza example raised here may be particularly applicable to humoral responses (ie, in Line 406 enhancing humoral responses, specifically) and how the referenced trial in neonates and

infants (another unique immune milieu may (or may not) translate to adolescents and adults, which are the focus of the remainder of this paper.

We thank the reviewer for providing feedback to improve the flow of this manuscript and have moved this portion into an earlier part of the manuscript, with inclusion of more discussion and references on this point in older age groups relevant to this study, which have been added to the revised manuscript in accordance to the reviewer's suggestion. We have also ended the manuscript with the stronger final sentence as suggested by the reviewer.

(Discussion, Page 16, lines 392-397)

Recipients of CoronaVac appear to mount lower antibody titres than BNT162b2. Our group and others had previously shown that intradermal administration of inactivated influenza vaccine can enhance humoral responses across different age groups.⁶⁰⁻⁶² Whether this vaccination route induces higher humoral responses against SARS-CoV-2 and its VOCs should be explored for CoronaVac, which is especially important for countries where mRNA vaccines are not available.

(Discussion, Page 18, lines 429-431)

Taken together, vaccination elicits robust immune responses and remains a key method for providing host protection against COVID-19 in adolescents.

(References, Page 33, references 61-62)

61. Chiu, S.S., Peiris, J.S., Chan, K.H., Wong, W.H. & Lau, Y.L. Immunogenicity and safety of intradermal influenza immunization at a reduced dose in healthy children. *Pediatrics* 119, 1076-1082 (2007).

62. Egunsola, O., et al. Immunogenicity and Safety of Reduced-Dose Intradermal vs Intramuscular Influenza Vaccines: A Systematic Review and Meta-analysis. *JAMA Netw Open* 4, e2035693 (2021).

Methods:

The non-blinded and non-randomized design is a serious consideration that could particularly undermine the veracity of their safety/reacto assessments and comparisons, particularly given the considerations already acknowledged by the authors with respect to the well-publicized safety profile of BNT162b2 among adolescents (e.g., pain, myocarditis). However, I think the authors adequately acknowledge this limitation of their study design and their reporting of the safety profiles and comparisons is fair.

We thank the reviewer for understanding the limitations of research studies, including ours, and that we have adequately acknowledged and presented.

Line 424: How was “known history of COVID-19” identified? This would be helpful to specify here, whether by potential participant self-report, baseline serologic testing, or otherwise.

Known history of COVID-19 was identified at screening and every visit encounter by self-report/clinical history. At the commencement of the study, the population infection rate in Hong Kong based on sero-epidemiology studies was only ~1% (unpublished data). Additionally, those with positive serological testing for N (those who received B or BB) or ORF8 IgG at those visits or baseline S-RBD IgG were excluded from final data analyses. These have been further clarified in the revised manuscript.

(Online Methods, Page 19, lines 447-448)

Potential participants needed to be healthy or in stable health condition, and those with known history of COVID-19 by self-report at any of the 3 study visits (also baseline S-RBD IgG positivity, or ORF8 IgG positivity at any visit, or N IgG positivity at any visit for B or BB), history of severe allergy, significant neuropsychiatric conditions, immunocompromised states, transfusion of blood products within 60 days, haemophilia, pregnancy or breastfeeding were excluded from this analysis.

Lines 588 and 593: The windows for post-dose 2 sampling timepoints are somewhat large—2-6 weeks in the “evaluable analysis population” or 1-8 weeks in the “expanded analysis population.” The need for wide windows in such trials is understandable, but can also significantly impact immunologic parameter measurements, particularly in such early post-vaccination weeks. Thus, the authors should clearly characterize this variability across the subgroups (eg, reporting mean post- dose 2 sampling time points and SD adolescents and adults in each vaccine regimen subgroup) and control for it in the comparative analysis of adolescent and adult titres, response rates, etc.

We thank this comment to check for the potential for wide windows for such trials that may influence variability across subgroups. The mean, standard deviations (SD) and differences (Diff) of time of blood sampling in days between visit 3 (post-dose 2) and dose 2 for adolescents and adults and for each vaccine are shown in the table below (the limit of tables and figures for this journal have been reached). The blood sampling time differences of <3 days in the evaluable groups and <10 days in the expanded groups were not expected to result in clinically meaningful differences in immunological responses, especially since the results of the immunological outcomes in the expanded analysis populations were consistent with the results of the evaluable analysis population. This point has been added to the revised manuscript.

	BNT162b2 Adolescents		BNT162b2 Adults		Diff	CoronaVac Adolescents		CoronaVac Adults		Diff
	Mean	SD	Mean	SD		Mean	SD	Mean	SD	
Evaluable	28.5	5.1	31.0	2.4	2.5	28.2	5.2	30.4	2.2	2.2
Expanded	29.0	7.0	33.2	4.2	4.2	28.4	10.4	38.0	2.8	9.6

Numbers are in days

(Results, Page 7, lines 174-175)

Demographic characteristics (Extended Data Table 1) and mean time of blood sampling (data not shown) were evenly distributed.

Reviewer: 2

This is a fantastic paper, beautifully and clearly written, and thoughtfully conducted. Have been waiting for data such as these, so it is wonderful to see. The policy relevance of the findings is well articulated in the discussion, which also appropriately addresses the question of omicron specific responses, a question which is begged throughout a reading of this paper. I do have (quite) a few questions though, but none of these detracts from the high quality and importance of this paper.

We thank the reviewer for informing us about the appreciation of our work.

line 129 I believe studies of T cell responses have also reported on preserved longevity of response. This is not mentioned in the introduction, but the authors do mention in in the discussion in passing with respect to HBV. I also wonder whether time since vaccine could be a factor for future analysis?

We are thankful to the reviewer for bringing up this important point. Indeed, we agree with the reviewer on this idea. This current manuscript presents the interim results only of a 3-year study, and we will continue working diligently and also await excitedly for the longer term follow-up immunogenicity outcome results as described in our Protocol and Statistical Analysis Plan.

We have added that this is an ongoing 3-year study in the Results section.

(Results, Page 7, lines 165-166)

This present interim analysis of our study that will track severe adverse events and immunogenicity outcomes over a 3-year period focused on healthy participants, consisting of 239 adolescents (11-17 years old, mean=14.0, SD=1.7) and 288 adult (18-67 years old, mean=47.5, SD=7.5) participants (total N=527) who received at least 1 dose of either BNT162b2 or CoronaVac.

line 221 The non-inferiority margin is 0.6. It is not stipulated if this is on a relative scale. The clinical or immunological justification for this margin is not given anywhere in the manuscript. And 0.6 seems rather a large margin. However in line 122 "GMR 0.48, 95% 0.42-0.54" is reported to fail non-inferiority, that is, there is evidence the result is inferior. But this is confusing since 0.48 is within the 0.6 margin. Could this be made clearer throughout?

Indeed, "in line 122 'GMR 0.48, 95% 0.42-0.54' is reported to fail non-inferiority, that is, there is evidence the result is inferior" is correct. The non-inferiority margin was predefined in the manuscript's attached study Protocol and Statistical Analysis Plan that were completed prior to the data analysis step, which was described in the Analysis populations subsection of the Online Methods. We appreciate the reviewer for requesting us to make clearer the non-inferiority margin on the relative scale of geometric mean ratio 0.6, and the line of 1 was considered no difference for the analysis subgroups. This has been clarified in the revised manuscript. We have also added references that were followed by this study for the reader's understanding of the confidence interval margins to fulfil non-inferiority, inferiority, non-inferiority and superiority, non-inferiority and inferiority, or inconclusive.

(Online Methods, Pages 26-27, lines 628-634)

The GMRs were reported with a two-sided 95% CI for testing the non-inferiority hypothesis at the criterion of 0.60, with the basis of WHO criterion of 0.67 balanced by a pragmatic approach to achieve rapid delivery of study results.^{75,77,78} Non-inferiority analyses were repeated in the expanded analysis populations, which also included more participants in a broader dosing and blood sampling intervals. Analysis subgroups were considered superior if the lower bound of the 95% CI for GMR with the comparator was >1 or inferior if the upper bound was <1.^{75,78}

(References, Page 34, references 75 and 77-78)

75. Liu, X., et al. Safety and immunogenicity of heterologous versus homologous prime-boost schedules with an adenoviral vectored and mRNA COVID-19 vaccine (Com-COV): a single-blind, randomised, non-inferiority trial. *Lancet* 398, 856-869 (2021).

77. World Health Organization. Guidelines on clinical evaluation of vaccines: regulatory expectations. <https://www.who.int/publications/m/item/WHO-TRS-1004-web-annex-9> (2017).

78. Bikdeli, B., et al. Noninferiority Designed Cardiovascular Trials in Highest-Impact Journals. *Circulation* 140, 379-389 (2019).

line 224 "were all significantly lower" seems no longer to be reporting on non-inferiority, but rather on the usual superiority (here inferiority) approach. Is this right?

We thank the reviewer in informing us that this sentence is not fully clear. We agree the manuscript flow can improve by first completing the presentation on the non-inferiority results and then present Extended Data Fig. 2a that shows the comparisons of humoral immunogenicity outcomes between groups by unpaired t test after natural log transformation (Online Methods, Page 25, lines 622-623).

(Results, Page 9, lines 229-232)

S-RBD IgG, sVNT, PRNT90 and PRNT50 were all significantly lower in adolescents than adults (GM OD450 1.96 vs 2.73, GM % inhibition 81.3% vs 94.9%, GM PRNT90 14.4 vs 64.6 and GM PRNT50 45.2 vs 259, respectively), all $P < 0.0001$ (Table 1a) (Extended Data Fig. 2a).

line 286/7 CoP and PRNT90. I am uncertain if the Khoury paper compared any assay in vaccinated vs the same assay in convalescent sera, or whether their curve was normed on PRNT50. Their text implies that might be the case. In any event, our authors performed both PRNT50 and PRNT90 (as stated in line 221), so could they address their use of PRNT90 for the CoP evaluation? (As an aside, I commend the authors for setting their own bar higher with the decision to evaluate PRNT90.)

We appreciate the reviewer for commending us and seeking clarity on this point. The Khoury paper was a meta-analysis on convalescent sera that included different studies, which did not use a single, definite neutralization cut-off. As the reviewer astutely pointed out, the reason for our choice of PRNT90 is its higher bar for the CoP evaluation. We had performed this analysis with PRNT50 (Adolescent CC: 52%; Adolescent B: 61%; Adolescent BB: 93%), which showed slightly higher VEs but were overall similar as for PRNT90. This note has been added into the Results.

(Results, Page 12, lines 298-299)

This analysis was also repeated for PRNT50, which yielded similar findings.

line 330 The argument in this sentence is unclear. The logical conclusion in the sentence (which is presented in the first half) does not follow from the premise in the sentence (presented in its second half). Why would preventing hospitalization in the young bring the pandemic to endemicity?

We thank the reviewer for notifying us that this sentence has insufficient logical flow and have modified in the revised manuscript, with a reference added to that provides a more comprehensive explanation to future readers on the rationale of this sentence.

(Discussion, Page 14, lines 340-342)

*Approval of vaccines in the younger age groups to prevent hospitalizations and reduce COVID-19 severity is key during and after the transition of the devastating pandemic to benign endemicity, which requires infections in all ages including children to be mild.*⁴¹

(References, Page 32, reference 41)

41. Antia, R. & Halloran, M.E. Transition to endemicity: Understanding COVID-19. *Immunity* 54, 2172-2176 (2021).

line 415-416 *The implied primary objective here is to compare across two vaccines in children. But in lines 157-158 the authors state the actual aim of noninferiority of responses in adolescents compares to adults. Seems that lines 415-6 should be revised.*

We thank the reviewer for alerting us that this point can be made clearer. There were 2 aims:

In this study,

- 1) we aimed to perform an immunobridging study showing the humoral and cellular immunogenicity in adolescents receiving 1 and 2 doses of BNT162b2 and 2 doses of CoronaVac are non-inferior to adults, especially to inform on the use of CoronaVac in children for which there are no efficacy and effectiveness data at the time of writing.
- 2) We compared various humoral and cellular response outcomes in adolescents to BNT162b2 and CoronaVac head-to-head, which are the top 2 most used COVID-19 vaccines in the world.

To keep consistency and clear understanding on the presentation of these two aims, we have split the one sentence into two sentences as follows:

(Discussion, Page 19, line 437)

Study Design. *Coronavirus disease-19 (COVID-19) Vaccination in Adolescents and Children (COVAC) is a registered clinical study (Department of Health, Hong Kong (HK), Clinical Trial Certificate 101894; clinicaltrials.gov NCT04800133) with a non-inferiority, non-blinded, non-randomized design aimed at establishing immunobridging for 2 COVID-19 vaccines, BNT162b2 (B) and CoronaVac (C), in children. Additionally, the study aimed to compare the reactogenicity and immunogenicity between the 2 vaccines in children.*

line 422 *Were the recruited adults related to the participating children? Line 569 says "comparing... of children with that of parents". If so, this would represent a convenience sample, but raise some issues regarding lack of independence between the groups, particularly with respect to presumed biological similarity which could bias towards a traditional null (though here toward non-inferiority), or at least imply a need for paired analyses (like McNemar tests where the authors used Fisher's exact test?) Actually I am quite unsure how one would deal with a non-inferiority framed question in the context of clustered non-independent responses. Please would the authors clarify?*

We thank the reviewer for raising this point for us to clarify. Our adult controls consisted of both parents/other family members and unrelated individuals (unrelated individuals: 46 out of

288 adults, or 16.0%, in the safety population; 33 out of 166, or 20.0%, in evaluable analysis population at visit 3; 39 out of 223, or 17.5%, in expanded analysis population at visit 3).

While it is possible that biological similarity between family members and children would make a non-inferior outcome more likely, we did not expect that to alter our results significantly. The inclusion of parents may allow us to dissect out the difference in vaccine responses due to age, reducing the impact of genetics as with unrelated adult controls. Our findings of non-inferiority for various antibody and T cell outcomes were also consistent with non-inferior and superior neutralizing antibody results from phase II trials for BNT162b2 and CoronaVac. As children-parent pairs were only part of our study populations, we utilized unpaired approaches for comparing responses between adolescents and adults.

This point has been clarified in the revised manuscript.

(Online Methods, Page 19, lines 444-445)

Participants. This analysis included 11- to 17-year-old children and ≥ 18 -year-old adults who were family members and unrelated individuals.

(Online Methods, Page 25, line 596)

For primary immunogenicity objectives, when comparing the peak geometric mean (GM) immunogenicity outcomes of children with that of adults, or between vaccine types, a sample size of 61 in each group would assure that a two-sided test with $\alpha=0.05$ has 99% power to detect an effect size with a Cohen's *d* value=0.78, or a difference of 0.51 after natural log transformation, between 2 groups and a standard deviation (SD) of 0.65 on the natural log scale within each group.

line 424 "known history of COVID-19" the authors don't explain how this was ascertained, was it by self report? or documentation? Was baseline serology done? Though in line 585 the authors report collecting clinical history, baseline S-RBD IgG negativity, N and ORF8 IgG negativity. And on line 271 the authors report on baseline T cell responses. So which of all these was used to establish this exclusion criterion?

Known history of COVID-19 was identified at screening and every visit encounter by self-report/clinical history. Additionally, those with positive serological testing for N or ORF8 IgG at those visits or baseline S-RBD IgG were excluded from final data analyses. These have been further clarified in the revised manuscript.

(Online Methods, Page 19, lines 447-449)

Potential participants needed to be healthy or in stable health condition, and those with known history of COVID-19 by self-report at any of the 3 study visits (also baseline S-RBD IgG positivity, or ORF8 IgG positivity at any visit, or N IgG positivity at any visit for B or BB), history of severe allergy, significant neuropsychiatric conditions, immunocompromised states, transfusion of blood products within 60 days, haemophilia, pregnancy or breastfeeding were excluded from this analysis.

lines 468, 505, 506, 525, 548x, 610: imputation of half the detectable lower limit was used widely in this work. How prevalent was non-detection? This should be reported, since it can introduce bias, especially for outcomes that are reported as fold- change (line 631).

We thank the reviewer for this question. Tables 1a, 1b, Extended Data Table 2 and Extended Data Table 3 list the prevalence of detection (“% positive (\geq LOD or LOQ or cut-off)).

(Results, lines regarding limit of detection and limit of quantification)

line 485 and 499. Was the pseudovirus that was used based on Wuhan parental strain? The authors mention in the discussion that omicron was not used. But please clarify the strain that was in fact used, in terms of its relation to dominant global variants.

We appreciate the reviewer for requesting for us to specify the strain used, which was based on the wild-type SARS-CoV-2 virus, and we have added this information into the revised manuscript.

(Online Methods, Page 22, line 511)

The sVNT was performed using 10 μ L of each serum, positive and negative controls, which were diluted at 1:10 and mixed with an equal volume HRP conjugated to the wild-type SARS-CoV-2 S-RBD) (6 ng).

(Online Methods, Page 22, lines 524-525)

Serial serum dilutions from 1:10 to at least 1:320 were incubated with ~30 plaque-forming units of the wild-type strain, SARS-CoV-2 BetaCoV/Hong Kong/VM20001061/2020 virus, for 1 hour at 37°C.

line 548 were any T cell functional assays performed, like proliferative responses or the like? Is this relevant?

We thank the reviewer for seeking clarification of this important point. Our assays were indeed T cell functional assays, which measured intracellular cytokine expression rather than proliferative responses. This information has been added in the revised manuscript.

(Online Methods, Page 24, lines 558-559)

For this T cell functional assay, thawed PBMCs were rested for 2 hours in 10% human AB serum supplemented RPMI medium.

line 569 - see line 422 above.

This has been addressed as above and revised accordingly in the revised manuscript.

line 570 two-tailed tests are mentioned. But this is unusual for a non-inferiority design.

We thank the reviewer for requesting an explanation for this method. One-sided 97.5% CI or two-sided 95% CI has been used in previous such studies and given that it would also be interesting to allow for a conservative estimation for superiority comparison, we chose the two-sided 95% CI approach. References to support this statistical method has been added to the revised manuscript.

(Online Methods, Page 25, line 601)

For primary immunogenicity objectives, when comparing the peak geometric mean (GM) immunogenicity outcomes of children with that of the adults, or between vaccine types, a sample

size of 61 in each group would assure that a two-sided test with $\alpha=0.05$ has 99% power to detect an effect size with a Cohen's d value=0.78, or a difference of 0.51 after natural log transformation, between 2 groups and a standard deviation (SD) of 0.65 on the natural log scale within each group.^{75,76}

(References, Page 34, references 75-76)

75. Liu, X., et al. Safety and immunogenicity of heterologous versus homologous prime-boost schedules with an adenoviral vectored and mRNA COVID-19 vaccine (Com-COV): a single-blind, randomised, non-inferiority trial. *Lancet* 398, 856-869 (2021).

76. D'Agostino, R.B., Sr., Massaro, J.M. & Sullivan, L.M. Non-inferiority trials: design concepts and issues - the encounters of academic consultants in statistics. *Stat Med* 22, 169-186 (2003).

line 577 When the authors state 7.5% do they mean 7.5 percentage points? Is a relative or absolute scale being used?

We thank the reviewer for improving this manuscript by requesting us to specify this more clearly, and we have amended to “percentage points” in the revised manuscript accordingly.

(Online Methods, Page 25, line 605)

For the proportion of participants with a positive result in immunogenicity outcomes or ARs, 110 adolescents would yield a 95% chance to detect the true value within ± 7.5 percentage points of the measured percentage, assuming a prevalence of 80%.

line 603-604 is a bit confusing, since it seems to imply a bioequivalence approach and not a non-inferiority approach.

We are very thankful for the reviewer to have informed us this explanation was a bit confusing, and we have amended to describe the first primary aim of this study was by non-inferiority and not equivalence approach. In addition to non-inferiority, we investigated whether the responses were inferior or superior, and therefore the possible outcomes included non-inferior, inferior, non-inferior and inferior, superior, non-inferior and superior, or inconclusive.

(Online Methods, Pages 26-27, lines 628-634)

The GMRs were reported with a two-sided 95% CI for testing the non-inferiority hypothesis at the criterion of 0.60, with the basis of WHO criterion of 0.67 balanced by a pragmatic approach to achieve rapid delivery of study results.^{75,77,78} Non-inferiority analyses were repeated in the expanded analysis populations, which also included more participants in a broader dosing and blood sampling intervals. Analysis subgroups were considered superior if the lower bound of the 95% CI for GMR with the comparator was >1 or inferior if the upper bound was <1 .^{75,78}

line 614 good to see a smaller alpha given multiple comparisons. But why was $P=0.01$ chosen? Arbitrarily or based on some specific consideration?

There were 10 variables for multiple comparisons. If Bonferroni adjustment is made, $P<0.005$ would be chosen as the significance level, which the authors, including the statistics team, deemed strict and can potentially increase type II error. To reduce type II error and for

simplicity, $P=0.01$ was arbitrarily chosen. It was noteworthy, though, that the majority of the significant correlation findings (15 out of 21) were $P<0.001$ and $P<0.0001$. Most importantly, whether the P is to be set as 0.05, 0.01 or 0.005 for the multiple comparisons adjustment, there was no correlation between the set of humoral immunogenicity and set of cellular immunogenicity outcome variables.

(Online Methods, Page 27, line 647)

Relationships between sVNT %inhibition and baseline variables such as age, sex and haematological parameters were explored by multiple linear regression post-dose 1 by vaccine type in the adolescent group only at $P=0.01$ arbitrarily for the multiple comparison adjustment.

line 634-635 Estimating VE based on immune response curves, though an important advance during the pandemic, is still no replacement for epidemiologically rigorous VE studies. So at very least the authors should address the caveats of this.

We agree with the reviewer and have added this caveat into the revised manuscript.

(Discussion, Page 17, lines 411-413)

It is necessary to note that estimating VEs based on immune response curves, though an important advancement during the pandemic, is still no replacement for epidemiologically rigorous VE studies.

Linguistic and other pedantry:

line 85 "in more adolescents than adults", unless denominators are equal, this statement is unclear. Perhaps the authors mean "more so in adolescents than in adults"

We appreciate the reviewer's feedback and have modified to better explain our meaning.

(Abstract, Page 3, line 84)

CC induced SARS-CoV-2 nucleocapsid (N) and N C-terminal domain seroconversion in a higher proportion of adolescents than adults.

line 104 "and effective" should be "and efficacious", since WHO EUL was on basis of RCT results.

We agree with the reviewer and have changed "effective" to "efficacious" accordingly.

(Introduction, Page 4, line 104)

Several vaccines against SARS-CoV-2, utilizing novel nucleoside-modified mRNA technologies and the conventional inactivated whole-virus platform, underwent an expeditious review process by the World Health Organization (WHO) and were deemed adequately safe and efficacious for emergency use in adults and older children only during the initial phase.^{5,6}

line 163 "of which 646 who provided consent"  "of whom 646 provided consent"

We thank with the reviewer and have changed "of which 646 who provided" to "of whom 646 provided consent" accordingly.

(Results, Page 7, lines 161-162)

Enrolment of study participants. A total 658 volunteers were screened, of whom 646 provided consent, consisting of 309 adolescents and 337 adults at dose 1, respectively, were enrolled between 27 April 2021 and 23 October 2021 (see Methods; Extended Data Fig. 1).

line 214 "and thus non-inferior and superior for adolescent CC" - this sentence is unclear. Was it cut off in error?

We thank the reviewer for informing us this and we have split this long sentence into two sentences. We have also added reference 78 (Bikdeli, B., *et al.* Noninferiority Designed Cardiovascular Trials in Highest-Impact Journals. *Circulation* **140**, 379-389 (2019), Figure 1) in the Online Methods for future readers to be clear about this result on satisfying the non-inferiority and superiority criterion.

(Online Methods, Page 9, lines 217-218)

Interestingly, for N IgG and N-CTD IgG, only a small proportion of adult CC seroconverted (52.4% and 28.6% vs adolescent CC of 98.4% and 92.2%, respectively, $P < 0.0001$ for both). Adolescent CC satisfied the non-inferior and superior criterion (N IgG: GMR 2.24, 95% CI 1.87-2.68; N-CTD IgG: GMR 2.27, 95% CI 1.82-2.82) (Fig. 1a_{iii}).

(References, Page 34, reference 78)

78. Bikdeli, B., *et al.* Noninferiority Designed Cardiovascular Trials in Highest-Impact Journals. *Circulation* **140**, 379-389 (2019).

line 226 "Despite satisfying non-inferiority for S-RBD IgG and sVNT, adolescent B was indeed inferior to adult BB on both tests" Suggest rephrase "Despite BB satisfying non-inferiority for S-RBD IgG and sVNT, adolescent B was indeed inferior to adult BB on both tests." (As a minor aside - failing non-inferiority does not make something "indeed inferior", it just makes it not non-inferior. Under Popperian principles, the null hypothesis can never be proven, it can only be rejected. But biologically stating this is inferior is indeed sensible, statistical technicalities notwithstanding.)

We thank the reviewer for asking us to more clearly describe these results. We have amended to the following and we have added reference 78 (Bikdeli, B., *et al.* Noninferiority Designed Cardiovascular Trials in Highest-Impact Journals. *Circulation* **140**, 379-389 (2019), Figure 1) in the Online Methods for future readers to be clear about this result on satisfying the non-inferiority and inferiority criterion.

(Online Methods, Page 9, lines 227-228)

Compared to adult BB, adolescent B satisfied the non-inferior and inferior criterion for S-RBD IgG and sVNT.

(References, Page 34, reference 78)

78. Bikdeli, B., *et al.* Noninferiority Designed Cardiovascular Trials in Highest-Impact Journals. *Circulation* **140**, 379-389 (2019).

line 525 The segment commences on avidity, then discussion Fc-R binding, the concludes with avidity again. Is this intentional? Should this be restructured, or does the final section refer also to the Fc-R binding assay?

We agree with the reviewer, and we have restructured these 3 sentences to fully describe the avidity and then Fc-R binding.

(Online Methods, Page 23, lines 545-548)

To assess antibody avidity, plates were washed 3 times with 8M Urea before incubation for 2 hours with IgG-HRP (1:5000; G8-185, BD). HRP was revealed by stabilized hydrogen peroxide and tetramethylbenzidine (R&D systems) for 20 minutes, stopped with 2N H₂SO₄ and analysed with an absorbance microplate reader at 450 nm wavelength (Tecan Life Sciences). The IgG avidity index was given by the ratio of the OD₄₅₀ values post-washing to pre-washing of the plates, which was only calculated when associated with a positive S IgG value and this was censored at 100%. To measure FcγRIIIa-binding antibodies, plates were instead coated with 500ng/mL S protein, incubated with HI serum at 1:50 dilution for 1 hour at 37°C and then with biotinylated FcγRIIIa-V158 developed in-house at 100 ng/ml for 1 hour at 37°C. Streptavidin-HRP (1:10,000, Pierce) was then used to detect presence of S specific FcγRIIIa-V158-binding antibodies. OD₄₅₀ values at or above the respective limits of detection (LODs) were considered positive, and values below were imputed as 0.5 of the LOD.

Reviewer: 3

The authors conducted a carefully designed head-to-head comparison of BNT162b2 and CoronaVac in adolescents with a remarkable and comprehensive characterization of immune response. In addition, adults receiving both vaccines were also enrolled to immunobridge between the two populations.

Higher anti-spike serological readouts were observed in adolescents receiving 2 doses of BNT162b2 than CoronaVac, while similar T cell responses were observed. The readouts in adolescents were not inferior to those in adults, supporting the use of CoronaVac in adolescent population in absence of efficacy data in regions where mRNA vaccines are not available. Furthermore, the authors analyzed immune readouts in adolescents receiving 1 dose of BNT162b2, which was inferior to those in adults receiving 2 doses of the same vaccine.

We thank the reviewer for this precise and accurate summary.

Two minor points:

1. A supplemental table listing the dosage of both vaccines in adolescents and adults will be very helpful.

The dosage of BNT162b2 for adolescents and adults was the same, which was 0.3 mL (equivalent to 30 µg of COVID-19 mRNA vaccine embedded in lipid nanoparticles), the vaccine manufacturer's current recommended dosage for ≥12 years old. Similarly, the dosage of CoronaVac for adolescents and adults was the same, which was 0.5 mL (600 SU of inactivated SARS-CoV-2 virus as antigen), the vaccine manufacturer's current recommended dosage for ≥3 years old. Since the limit of tables and figures for this journal has been reached, we have added this information into the main text in the revised manuscript.

(Online Methods, Page 20, lines 459-461)

The dosages of BNT162b2 and CoronaVac were 0.3 mL (30 µg of COVID-19 mRNA vaccine embedded in lipid nanoparticles) and 0.5 mL (600 SU, equivalent to 3 µg, of inactivated SARS-CoV-2 virus as antigen), respectively.

2. A more detailed writeup of the PRNT assay, or a reference (if there is one) will be helpful.

We are thankful to the reviewer for requesting more information about the PRNT assay. We have added references into the paragraph describing the PRNT in the Online Methods section of the revised manuscript.

(Online Methods, Page 22, line 523)

The PRNT was performed in duplicate using culture plates (Techno Plastic Products AG, Trasadingen, Switzerland) in a biosafety level 3 facility.^{38,39,71}

(References, Pages 32 and 34, references 38-39 and 71)

38. Lau, E.H.Y., et al. Neutralizing antibody titres in SARS-CoV-2 infections. *Nat Commun* **12**, 63 (2021).

39. Lau, E.H., et al. Long-term persistence of SARS-CoV-2 neutralizing antibody responses after infection and estimates of the duration of protection. *EClinicalMedicine* **41**, 101174 (2021).

71. Perera, R.A., et al. Serological assays for severe acute respiratory syndrome coronavirus 2 (SARS-CoV-2), March 2020. *Euro Surveill* **25**(2020).